# Analysis of News Media-Reported Snakebite Envenoming in Nepal during 2010–2022

**Deb P. Pandey**[1]*, **Narayan B. Thapa**[2]

**1** Department of Veterinary Microbiology and Parasitology, Agriculture and Forestry University, Rampur, Chitwan, Bagmati Province, Nepal, **2** Department of Pediatrics, Bharatpur Hospital, Bharatpur, Chitwan, Bagmati Province, Nepal

* debpandey@gmail.com

## Abstract

### Background

Snakebite envenoming is a well-known medical emergency in the Terai of Nepal in particular. However, there is an epidemiological knowledge gap. The news media data available online provide substantial information on envenomings. Assessing this information can be a pristine approach for understanding snakebite epidemiology and conducting knowledge-based interventions. We firstly analyzed news media-reported quantitative information on conditions under which bites occur, treatment-seeking behavior of victims, and outcomes of snakebite envenomings in Nepal.

### Methodology/Principal findings

We analyzed 308 Nepalese snakebite envenomed cases reported in 199 news media articles published between 2010 and 2022 using descriptive statistics, Wilcoxon, and Chi-square tests to know why and how victims were bitten, their treatment-seeking behavior, and the outcomes. These envenomated cases known with substantial information represented 48 districts (mostly located in the Terai region) of Nepal. These envenomings mostly occurred in residential areas affecting children. Generally, envenomings among males and females were not significantly different. But, in residential areas, females were more envenomed than males. Further, victims' extremities were often exposed to venomous snakebites while their active status and these episodes often occurred at night while victims were passive during snakebites indoors and immediate surroundings of houses. Snakebite deaths were less among referred than non-referred cases, males than females, and while active than passive conditions of victims.

### Conclusion/Significance

The most of reported envenomed patients were children, and most envenomings were due to cobra bites. Consultation with traditional healers complicated snakebite management. In most cases, deaths that occur without medical interventions are a severe snakebite consequence in Nepal. Further, several deaths in urban areas and mountains and higher hills of Nepal suggest immediate need of snakebite management interventions in the most affected

**Data Availability Statement:** The data that support the findings of this study are available as supplementary tables (S1 through S3 Tables).

Additionally, we uploaded data file used to analyses (S1 Data).

**Funding:** The author(s) received no specific funding for this work.

**Competing interests:** The authors have declared that no competing interests exist.

districts. Therefore, there is an urgent need to immediately admit Nepalese snakebite victims to nearby snakebite treatment centers without adopting non-recommended prehospital interventions. The strategies for preventing snakebite and controlling venom effects should also include hilly and mountain districts where snakebite-associated deaths are reported.

## Author summary

Worldwide, 1.8 to 2.7 million snakebite envenomings occur annually resulting in 81,000 to 138,000 deaths. Annually, 1,000 to 3,225 deaths occur in Nepal. However, there is an epidemiological knowledge gap. Since news media data are typically available in near real-time, the news media data can be analyzed to understand the epidemiological situations in Nepal like poor resource country. Therefore, we analyzed news media-reported snakebite envenomings that occurred in the last 13 years. The median age of envenomed patients was 19 y. People were envenomated mainly during natural human activities such as sleeping. Cobras, kraits, Russell's Viper, Mountain Pitviper, and Green Pitvipers were involved in snakebites. Overall, three-fourth of victims who consulted traditional healers died. Forty-four cases being referred to higher centers were died on the way to higher healthcare centers or in healthcare institutions during treatment. Dependency on traditional snakebite healers, the long distance between snakebite localities and snakebite treatment centers, and snakebite occurred at night were the major barriers to accessing healthcare facilities by people bitten by snakes in Nepal. In this country, people inhabiting 48 districts are at risk of venomous snakebites.

## Introduction

Globally, snakebite results in 1.8 to 2.7 million envenomings [1,2] and 81,000 to 138,000 deaths annually [3]. The highest incidence of snakebite envenomings and deaths occurs in South Asia [1], particularly in India and Pakistan [4] where it has considerable social and economic impacts. In Nepal, 20,000 to 37,661 people are bitten by snakes resulting in 1,000 to 3,225 deaths annually [5,6]. However, there is an epidemiological knowledge gap due to inconsistent and incomplete hospital medical records of admitted snakebite cases [7,8], and limitations exist in community-based snakebite studies [9]. One recent community-based study in the Terai region of Nepal reported the fatality rate of snakebite envenoming to be 22.4 per 100,000 [6] which is over five times of the recent estimate of the fatality rate for India [4]. During the cross-sectional survey between 30th November 2018 and 7th May 2019 in the 23 districts of Terai region of Nepal [6], the authors excluded towns and cities where snakebite envenomings and associated mortalities are frequently reported [10–12], but proportion of envenomings and human population is less. The inclusion of rural areas having high snakebite incidence but less human population caused higher proportion of envenomings and associated consequences compared to the urban areas. Hence, in Terai of Nepal, there might be a cause of higher fatality rate than that is reported in India. At least, this exclusion failed to represent snakebite envenoming in tropical urban areas of Nepal's Terai.

The news media data available online provide useful information on envenomated cases even from towns and cities (being treated in healthcare centers of Nepal, consulting traditional healers, and adopting other domestic remedies [13]), with addresses of snakebite locality, demographics, treatment-seeking behavior, and outcomes. Such information can be used to

understand the epidemiological situation. Further, these data are useful to design the regionally or nationally representative snakebite study by including rural, semi-urban, and urban areas at the risk of snakebite to determine actual burden due to snakebite envenomings.

Nepal has significantly progressed in print to online news media and journalism sectors since the restoration of multiparty democracy in 1990 [14] although the first print news media–Gorkhapatra was published in 1858 [15]. The number of outlets and news media coverage has rapidly increased across the country. At least 863 news media publish news regularly in the Nepali language predominantly, followed by English and indigenous languages [16].

Assessing news media-reported snakebite case reports can be a novel approach for examining the demographics and circumstances of envenoming, treatment-seeking behaviour of the snakebite patients, and outcomes in Nepal having inadequate healthcare facilities, although this approach has advantages and some limitations [17]. Since data from news media are typically available in near real-time and provide earlier estimates of epidemic issues [18], the trend of snakebite envenomings and associated consequences can be understood by analyzing media-reported snakebites. Therefore, analyzing news-media reported snakebite envenomings is essential to understand snakebite epidemiology and conduct knowledge-based interventions [19].

Although our news media-based data set is a subset of all envenomed cases known from healthcare facilities and out-of-hospitals and likely to represent a skewed sample due to various biases in reporting, this data set analyzes out-of-hospital cases and deaths, which do not appear in a hospital-based data set. Further, this subset of cases also includes substantial details of snakebite events from urban to rural areas in the Terai, hills, and the mountains of Nepal, which are often absent from hospital- and community-based studies. However, there has yet to be an analysis of media-reported snakebite cases in this country. Therefore, assessing media-reported snakebite envenomings with substantial information can be a new approach for making these under-studied snakebite envenomings a critical issue because envenomings are extending vertically towards hills and mountains as well.

Herein, we analyzed news media-reported snakebite envenomings that occurred in Nepal between 2010 and 2022 to determine the distribution of snakebite envenomings in districts and along the altitudes and the climatic zones and understand the demographics (age, sex, and occupation of envenomed patients), circumstances (locality/places of snakebite envenomings, victims' activities when bitten, bitten body parts, time, month, and season of snakebite envenomings, and snakes responsible for envenomings), victims' time to hospital arrival and treatment-seeking behavior, and outcomes. Further, we tested hypothesis to know whether snakebite envenomings occurred according to the demographics and circumstances and to identify any association between multiple variables (for example: human habitations and seasons, activeness of victims and sex, etc.).

## Methods

### Study site and population

Nepal has diversified lands extending 60–8,848 m above sea level (asl). Nepal's Terai (i.e. the plains running parallel to the lower ranges of the southern Himalayas of Nepal) is characterized by a hot tropical climate, the mid-elevation Chure hills by a mild subtropical climate, and the high-elevation Mahabharat ranges by temperate and subalpine climates (Fig 1). The Terai is the northernmost Ganges plains extending south to north by 30–40 km at 60–200 m asl occupying 4% of the total area of the country and the Chure hill ranges is Siwalik region extending at 200–1,500 m asl occupying roughly 13% of the country area. The Mahabharat ranges are the middle hills extending east to west continuously from 1,500–4,000 m asl

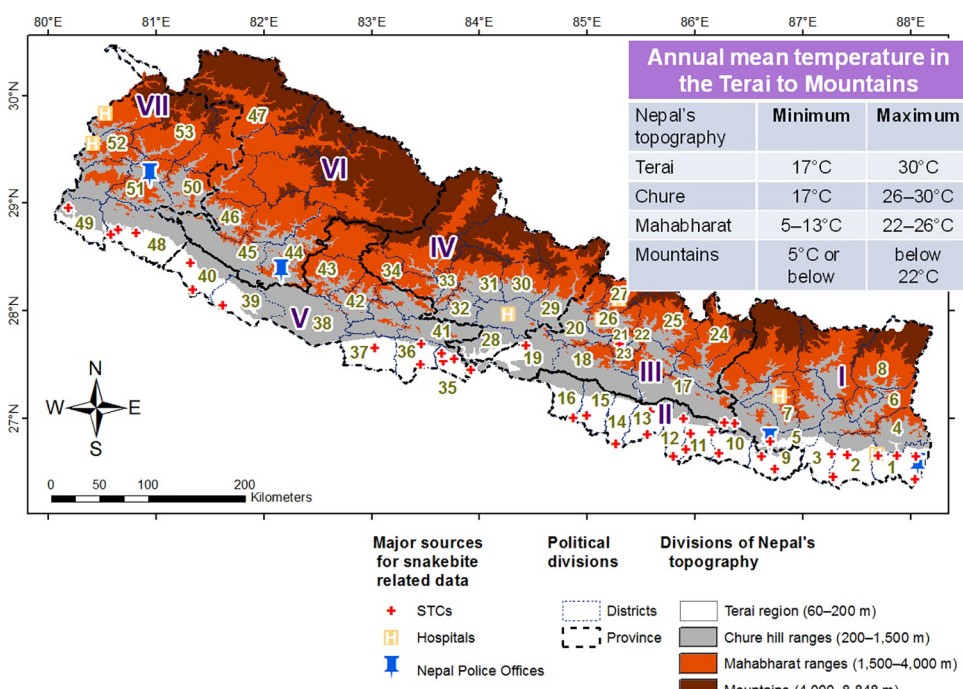

**Fig 1. Divisions of Nepal's topography and geographic locations of the major sources (i.e., the "STCs" which stands for Snakebite Treatment Centers; hospitals where antivenom supply was unclear, and Nepal Police Offices, S2 Table) for news media-reported snakebites included in this study.** Those news (S1 Table) reported snakebites from 53 districts that are displayed in numerals 1 through 53 from the eastern to far-western Nepal below:

| Provinces | Districts |
|---|---|
| I: Koshi Province | 1: Jhapa; 2: Morang; 3: Sunsari; 4: Ilam; 5: Udayapur; 6: Panchthar; 7: Khotang; 8: Taplejung |
| II: Madesh Province | 9: Saptari; 10: Siraha; 11: Dhanusha; 12: Mahottari; 13: Sarlahi; 14: Rautahat; 15: Bara; 16: Parsa |
| III: Bagmati Province | 17: Sindhuli; 18: Makawanpur; 19: Chitwan; 20: Dhadhing; 21: Kathmandu; 22: Bhaktapur; 23: Lalitpur; 24: Dolakha; 25: Sindhupalchowk; 26: Nuwakot; 27: Rasuwa |
| IV: Gandaki Province | 28: Nawalpur (aka Nawalparasi East); 29: Gorkha; 30: Lamjung; 31: Kaski; 32: Syangja; 33: Parbat; 34: Baglung |
| V: Lumbini Province | 35: Parasi (aka Nawalparasi West); 36: Rupandehi; 37: Kapilvastu; 38: Dang; 39: Banke; 40: Bardiya; 41: Palpa; 42: Pyuthan; 43: Rolpa |
| VI: Karnali Province | 44: Salyan; 45: Surkhet; 46: Dailekh; 47: Humla |
| VII: Sudurpaschim Province | 48: Kailali; 49: Kanchanpur; 50: Achham; 51: Doti; 52: Baitadi; 53: Bajhang |

[The first author of this study created this map in ArcGIS 10.1. The source of the basemap shapefile onto which data has been plotted was used from an openly available source (https://gadm.org/data.html)].

occupying 68% and mountains are highlands extending above 4,000–8,848 m asl occupying 15% of the total area of the country [20] (Fig 1).

Nepal is traversed by three main river systems: the Koshi in eastern, the Narayani in central, and the Karnali in western Nepal. During the monsoon season large flooding occurs in these river systems. This increases human-snake confrontations resulting in an increase of snakebites [21]. Therefore, we defined seasons according to the rainfall pattern as: the pre-monsoon (March–May), the monsoon (June–September), and the winter (October–February) to understand the seasonal influence on snakebite envenomings.

In Nepal, at least 18 species of medically relevant venomous snakes are distributed [22,23] within a small area (147,181 km$^2$ representing about 0.1% of the global landmass) extending east to west by 885–900 km and north to the Himalaya and to its Terai in the south by 130–260 km, respectively [20,24]. Elapid snakes, particularly the Spectacled Cobra (*Naja naja*) and the Common Krait (*Bungarus caeruleus*), cause most of the mortality in Terai of Nepal [25]. Viperid snakes, particularly, pitvipers are the cause of snakebite morbidity mostly in hills and mountains up to 5,000 m asl [22]. Russell's Vipers (*Daboia russelii*) are distributed mainly in Terai region of Nepal where their venom effects are associated with chronic wounds (which sometimes, may lead to amputation of the bitten body part) [10,22,26]. Krait species are active at night, cobras at dawn and dusks, and vipers during day and night. Kraits (*B. caeruleus*, *B. niger*, and *B. lividus*) and cobras (*N. naja*) are perianthropic species [10,22,27–29]. Humans and these snakes interact in residential areas often. Similar interactions of Russell's Viper (*D. russelii*) are reported often from agricultural lands in this country [10]. Pitvipers (mostly *Trimeresurus* spp. and *Ovophis monticola*) encounter with humans often in bushy areas in the hills and mountains [30,31].

According to Nepal Population Census 2021, total population of Nepal is 29,192,480. Total population of 48 districts (Table 1), from where narration of snakebites was reported in news (S1 Table), is 21,620,037 i.e., 74% of national population (Table 1). Since 2015 Federal Democratic Republic of Nepal is divided into seven provinces (the first level administrative units), 77 districts (the second level units), 753 local bodies (the third level units: six metropolitan cities, 11 sub-metropolitan cities, 276 municipalities, and 460 rural municipalities), and 6,443 wards (the last level units) (cited in: https://en.wikipedia.org/wiki/Village_development_committee_(Nepal), accessed in 17 July 2023). Formerly, Nepal was divided into five development regions (the first level units), 14 zones (the second level units), 75 districts (the third level units), and 58 municipalities and 3,157 VDCs (the fourth level units). Each of VDCs was divided into–on the average–nine wards (the last level units) depending on the population.

## Data sources and sample size

During August–December 2021 (while searching for traditional snakebite healers for a study [13]) and from January 2022 to the 11th of January 2023, we retrospectively searched international, national, and local news outlets including radio and television channels (e.g., BBC, Alzajira) in some extent for snakebites that occurred between January 2010 and December 2022 in Nepal. We searched at google.com using specific keys, "All filters and News", and customized date range and relevance options under the "Tool" given in https://www.google.com/. We also tracked the original web links for snakebite news (if any) posted on Facebook (we did not search other social media). We used snakebite-related terms "snakebite," "snake bite," "bitten by a snake," "Krait bite," "Cobra bite," and "Viper bite" in association with "Nepal" and the name of snakebite-prone districts listed by Bista et al. [32] and name of hospitals supplied with antivenom in English as well as Nepali language to find Nepalese snakebite related newspaper articles. Nepali unicode software (https://nepali-unicode-romanized.software.informer.com/download/) was applied while using respective keys in Nepali. We followed the additional news links for snakebite cases in the initially selected newspaper if links for other snakebite issues from Nepal were provided. Additionally, we used the search option for old archived news whenever available. Also, we retrospectively searched printed newspapers. These data were primarily based on Nepalese hospitals (Hs) /snakebite treatment centers (STCs) and Nepal Police Offices (NPOs). In Hs, snakebite cases accessed, but antivenom availability was unclear. The STCs are healthcare facilities dedicated to snakebite treatment where Nepal Government supplies antivenom free of costs for the use by Nepali citizen. The STCs included

Table 1. Distribution of envenomings (N = 308) and associated deaths with known demographic features, circumstances of bites, and treatment seeking behavior of the victims.

| Codes | | Districts (the numeral in the parenthesis represents respective district's human population as mentioned in Nepal Population Census 2021; 29 districts having antivenom treatment facilities† are in bold; ¶envenomings known without district distinctly = 2: *one envenomings in Terai areas and one envenoming resulting in death known from Nawalparasi i.e., either Nawalpur or Parasi District) | Incidence data | | Year ranges (within which cases were reported) |
|---|---|---|---|---|---|
| DC | PC | | Envenomings | Deaths | |
| | | | n = 306¶ | n = 228* | |
| 9 | II | **Saptari** (639284) | 39 | 23 | 2012, 2015, 2017–22 |
| 12 | II | **Mahottari** (627580) | 25 | 24 | 2015–18, 2020–22 |
| 14 | II | **Rautahat** (686723) | 23 | 11 | 2014–19, 2021 |
| 38 | V | **Dang** (552583) | 23 | 19 | 2015, 2017–22 |
| 49 | VII | **Kanchanpur** (451248) | 21 | 15 | 2010, 2015–17, 2020–22 |
| 1 | I | **Jhapa** (812650) | 16 | 15 | 2013, 2015–16, 2018–19, 2021–22 |
| 2 | I | **Morang** (965370) | 12 | 8 | 2015, 2017–19, 2021–22 |
| 5 | I | **Udayapur** (317532) | 12 | 12 | 2010, 2013, 2016, 2018–20 |
| 8 | I | Taplejung (127461) | 11 | 8 | 2012, 2015, 2019–21 |
| 15 | II | **Bara** (687708) | 8 | 6 | 2010, 2013–14, 2016, 2020–21 |
| 40 | V | **Bardiya** (426576) | 8 | 5 | 2012, 2015–16, 2018, 2020, 2022 |
| 48 | VII | **Kailali** (775709) | 8 | 6 | 2010, 2013–14, 2018–20 |
| 10 | II | **Siraha** (637328) | 6 | 6 | 2017–18, 2020, 2022 |
| 17 | III | **Sindhuli** (296192) | 6 | 3 | 2014, 2017–19, 2021–22 |
| 4 | I | Ilam (290254) | 5 | 5 | 2011, 2017, 2019 |
| 13 | II | **Sarlahi** (769729) | 5 | 5 | 2015, 2019, 2022 |
| 28 | IV | **Nawalpur (aka Nawalparasi East;** 311604) | 5 | 5 | 2016, 2021 |
| 35 | V | **Parasi (aka Nawalparasi West;** 331904) | 5 | 3 | 2015, 2021–22 |
| 39 | V | **Banke** (491313) | 5 | 2 | 2016, 2018, 2020 |
| 51 | VII | **Doti** (211746) | 5 | 5 | 2015, 2019–20 |
| 52 | VII | Baitadi (250898) | 5 | 4 | 2019–22 |
| 3 | I | **Sunsari** (763497) | 4 | 4 | 2015, 2019–20 |
| 19 | III | **Chitwan** (579984) | 4 | 2 | 2018, 2020 |
| 45 | VI | **Surkhet** (350804) | 4 | 3 | 2018, 2020, 2022 |
| 50 | VII | **Achham** (257477) | 4 | 4 | 2017, 2018, 2021–22 |
| 20 | III | Dhadhing (336067) | 3 | 1 | 2016, 2022 |
| 36 | V | **Rupandehi** (880196) | 3 | 3 | 2015, 2020, 2022 |
| 44 | VI | Salyan (242444) | 3 | 3 | 2017, 2022 |
| 47 | VI | Humla (50858) | 3 | 3 | 2018–19 |
| 16 | II | **Parsa** (601017) | 2 | 1 | 2015, 2020 |
| 22 | III | Bhaktapur (304651) | 2 | – | 2015 |
| 31 | IV | **Kaski** (492098) | 2 | – | 2020–21 |
| 41 | V | **Palpa** (261180) | 2 | 2 | 2014, 2018 |
| 46 | VI | Dailekh (261770) | 2 | 2 | 2017 |
| 53 | VII | Bajhang (195159) | 2 | 2 | 2012, 2019 |
| 6 | I | Panchthar (191817) | 1 | – | 2017 |
| 11 | II | **Dhanusha** (754777) | 1 | 1 | 2020 |
| 18 | III | Makawanpur (420477) | 1 | – | 2022 |
| 21 | III | **Kathmandu** (1744240) | 1 | 1 | 2016 |
| 24 | III | Dolakha (186557) | 1 | – | 2022 |
| 25 | III | Sindhupalchowk (287798) | 1 | – | 2016 |
| 27 | III | Rasuwa (43300) | 1 | 1 | 2015 |

*(Continued)*

**Table 1.** (Continued)

| Codes | | Districts (the numeral in the parenthesis represents respective district's human population as mentioned in Nepal Population Census 2021; 29 districts having antivenom treatment facilities† are in bold; ¶envenomings known without district distinctly = 2: *one envenomings in Terai areas and one envenoming resulting in death known from Nawalparasi i.e., either Nawalpur or Parasi District) | Incidence data | | Year ranges (within which cases were reported) |
|---|---|---|---|---|---|
| DC | PC | | Envenomings | Deaths | |
| | | | n = 306¶ | n = 228* | |
| 29 | IV | Gorkha (271061) | 1 | – | 2019 |
| 30 | IV | Lamjung (167724) | 1 | 1 | 2014 |
| 32 | IV | Syangja (289148) | 1 | 1 | 2022 |
| 37 | V | **Kapilvastu** (571936) | 1 | 1 | 2020 |
| 42 | V | Pyuthan (228102) | 1 | 1 | 2018 |
| 43 | V | Rolpa (224506) | 1 | 1 | 2022 |

**Abbreviations and symbols: aka**:

**DC:** district code as displayed in Fig 1 (envenomings were not reported from three districts: 7. Khotang, 23. Lalitpur, 26. Nuwakot. Only incidence of evenomings and deaths were mentioned without detailed information in cases from 33. Parbat and 34. Baglung. So, we did not include these two districts' data for the analyses.)

**PC:** Province codes in roman numerals I through VII (I: Koshi Province; II: Madesh Province; III: Bagmati Province; IV: Gandaki Province; V: Lumbini Province; VI: Karnali Province; VII: Sudurpaschim Province)

† Cited in: Annual report 2077/78 (2020/2021); p.198–99. Edited by Department of Health Services. Kathmandu: Annual Report published by Goverment of Nepal, Ministry of Health and Population, Department of Health Services. Available at: https://dohs.gov.np/annual-report-fy-2077-78-2019-20

healthcare centers, hospitals, and institutions led by Nepal Army and Nepal Red Cross Society, too.

To estimate the appropriate sample size (i.e., the expected number of envenomings from the aforementioned population of Nepal), we used Yamane formula [33]: $n = \frac{N}{1+Nd^2}$ where, n = sample size, N = population size, d = margin of error (aka precision) which was 10% of average prevalence of snakebite envenomings (i.e., 58%) reported in different studies conducted in hospitals of western Nepal (54.3% and 69%, respectively, [34,35]), a tertiary care center in the eastern Terai (88%, [36]), and communities of southcentral (42%, [37]) and southeastern Nepal (52%, [38]]). The estimated sample size was 297. As our samples (i.e., snakebite envenomings) were taken randomly from the media reports and represented 62% of total districts and 74% of total population of this country, it was adequate to test the hypothesis of the research objectives.

## Inclusion and exclusion criteria

We included news media reporting Nepalese snakebite cases that occurred between January 2010 and December 2022. Eligible reports were one of two types: individual case reports or incidence and case reports. For both types, only those with identifiable location were included. These included: certain healthcare facilities (hospitals or STCs), NPOs, or geographic locations (addresses, districts, or provinces where snakebite occurred). The location of the victim(s) was required to be in Nepal, even if the case was referred to the Indian healthcare system. In addition to the date and location, case reports were also included if they contained demographics, circumstances of the bite, prehospital care, and outcome.

We excluded only incidence data with known date, location, and counts of envenomings, deaths, bites without envenomings, and undetermined snakebites. We excluded news media-reported snakebites that occurred out of Nepal (herein, Indian snakebite patients) receiving treatment from Nepalese healthcare systems across the Terai of Nepal bordering India. We excluded news media describing only STCs or antivenom shortage, venoms, antivenom production, or expert opinions or recommendations on snakes and/or snakebites, the

"Naagpanchami" (a snake festival celebrated by Hindus), familiarizing snake rescuers, involvement of traditional snakebite healers (TSHs) in snakebite care. We excluded journal articles describing snakebites, personal websites, and blogs. We also excluded Facebook describing snakebites without original webpage links of the news media. Further, we excluded the duplicate news reports of the same case reports.

## Data collection and management

We amassed information such as the districts where bites occurred, age, sex, and occupation of snakebite cases, circumstances of the snakebites (i.e., time, month, and year of snakebites, localities where snakebites occurred, activity of the victims at the time of the bites, body parts bitten, and types of snakes involved in bites), prehospital intervention (the duration to reach the STCs/hospital [DOH]) and treatment-seeking behavior (i.e., carried to TSHs and/or healthcare facilities), and outcomes (discharged with complete recovery or disability, referred to the higher center, deaths (including places where deaths occurred such as personal homes, TSHs' homes, hospitals, on the way to hospitals), and length of time between the snakebite and declaration of deaths [LOD]).

We managed data using MS Excel. We crosschecked patients' names and demographics (including addresses where snakebites occurred, sex, and age), and assigned alphanumeric codes to avoid multiple entries of snakebite cases. We used "https://www.hamropatro.com/date-converter" to convert Nepali dates of snakebites and publications of news into respective English dates. To avoid duplication of snakebite incidence, we crosschecked the year and month of reports with the name of sources from where journalists extracted snakebite data. However, we extracted data from multiple news to make more comprehensive database of case reports if news articles had duplication in news titles with additional information about the cases (S1 Table).

## Data analysis

We analyzed only envenomated case reports with substantial information. For the analyses of demographics, circumstances, prehospital interventions, and outcomes of 308 snakebite envenomings, we analyzed envenomed cases with known geographic locations (i.e., village, town, municipality, and district) where snakebite occurred or STCs which provided case reports, sex, age, and age or age group without defining the sex of patients. We included envenomed cases with known: i) geographic locations and ii) sex (when we found information on everything else including sex but did not find age, we included the case for the analyses) and/or age/age group without sex (when we found information on everything else including age or age group but did not find sex, we included the cases for the analyses). However, when we found information on everything else but did not find sex or age/age group, we excluded the cases from analyses. Except for an envenomed case reported from the Terai of Nepal, 307 cases were provided with addresses for which we generated latitude, longitude, and elevation of localities where envenomings occurred using Google Earth Pro. We grouped victims aged 1–17 years (y) into children, 18–40 y adults, and 41 y and above into elders. We grouped the cases with "child, i.e., *Balak* in Nepali" or "young or youth, i.e., *Yuwa* in Nepali" without specific age into children and adults, respectively. Further, to know the occurrence and trend of envenoming at age intervals by 10 y and sex, we developed a line graphs.

To analyze the circumstances of envenoming, we categorized the place of bites that occurred as either in or out of human habitations (i.e., buildings including their yard/backyard, barns, and sheds). Further, we defined victim activity based on the description of what they were doing when actually bitten. At the same time, snakebites occurred during victim's

inactive status (such as sleeping or resting) and active status (e.g., walking, playing, among other activities). Also, we analyzed the bitten body parts under the reported circumstances. To interpret the reported time and seasonal patterns of snakebite, we grouped the bite time as early morning (if the reported bites occurred during 03:00–04:59 h), morning (05:00–09:59 h), day (10:00–16:59 h), evening (17:00–19:59 h), and night (20:00–02:59 h) time bites and bite season as: pre-monsoon, monsoon, and winter bites. Whenever the periods were mentioned without the specific time of the bites, we placed them into the respective categories as such. For the reports without a defined month of bites, we considered a month of news published as the month of the actual snakebites. It was because news media published snakebite events quickly to draw their readers' attention. Then, we analysed the risk periods and months/seasons of snakebites using frequency distributions to understand the temporal and seasonal influence on snakebites.

Whenever a newspaper reported the type of snake responsible for biting, we expected journalists to correctly identify the snake based on photographs of the responsible snakes taken by victims or their family members or dead snakes brought to healthcare facilities by consulting with snake experts. Based on available guidelines to identify snakes [22,39], we identified English, genus, or species names for the given vernacular names of snakes involved in envenoming. We divided these snakes into elapids [i.e., *Bungarus* spp. (kraits), *Naja* spp. (cobras)], viperids [i.e., *Daboia russelii* (Russell's Viper), *Trimeresurus* spp. (Green Pitvipers), *Ovophis monticola* (Mountain Pitviper)], and unidentified venomous snakes.

We analyzed prehospital interventions by assessing time to hospital arrival [DOH; it was converted into hour (h)] and those following TSHs and/or modern healthcare institutions for snakebite treatment. Outcomes were measured in terms of the number of envenomed cases who survived, died, and were under treatment. Further, we measured length of time between the snakebite and declaration of deaths [LOD].

Continuous data (herein, altitudinal ranges where envenomings occurred, age, DOH, and LOD) for those envenomated cases with substantial information were not normally distributed. We used box plots, the Grubbs test, and interquartile ranges (IQR) to identify any outliers in the dataset and histogram and the Shapiro-Wilk test to examine the normality of data distribution. Hence, we presented continuous data as medians, IQR, and ranges. We measured 95% confidence interval (CI) of elevation of areas where envenoming occurred and age, DOH, and LOD of patients using two-tailed Wilcoxon signed rank test with continuity correction in which median (instead of mean) was used. This CI was used to estimate the location of expected population median within the CI. We analyzed categorical data as proportions with percentages from the total eligible envenomed cases (for example, 308) but not from the article numbers (i.e., 533, 296, and 199, Fig 2). Following complete case analysis [40], we excluded missing values (i.e., NA). This reduced the cases from a total of 308 envenomed cases while analysing percentage in some cases. We used these percentages next to each of mentioned number of patients. We defined the absence of values for the aforementioned variables in news articles as "Not available" or "NA". These missing of values were completely at random and unrelated to any of the variables involved in our analysis. Therefore, we adopted a complete case analysis [40] and excluded all missing values (i.e., NA) on variables involved in the inferential statistical analysis and figures.

We compared snakebite across age-classes, sexes, occupations, snakebite locations, activities performed by victims while snakebite, bitten body parts, temporal patterns (time, months, and seasons) of snakebite envenomings, and types of snakes involved in bites using Pearson's Chi-squared test for goodness of fit (PCTGF). To compare the categorical variables, we used Fisher's exact test (FET) when there was at least one cell in the contingency table of the expected frequencies of observations was below five (variables: activeness and occupations of victims,

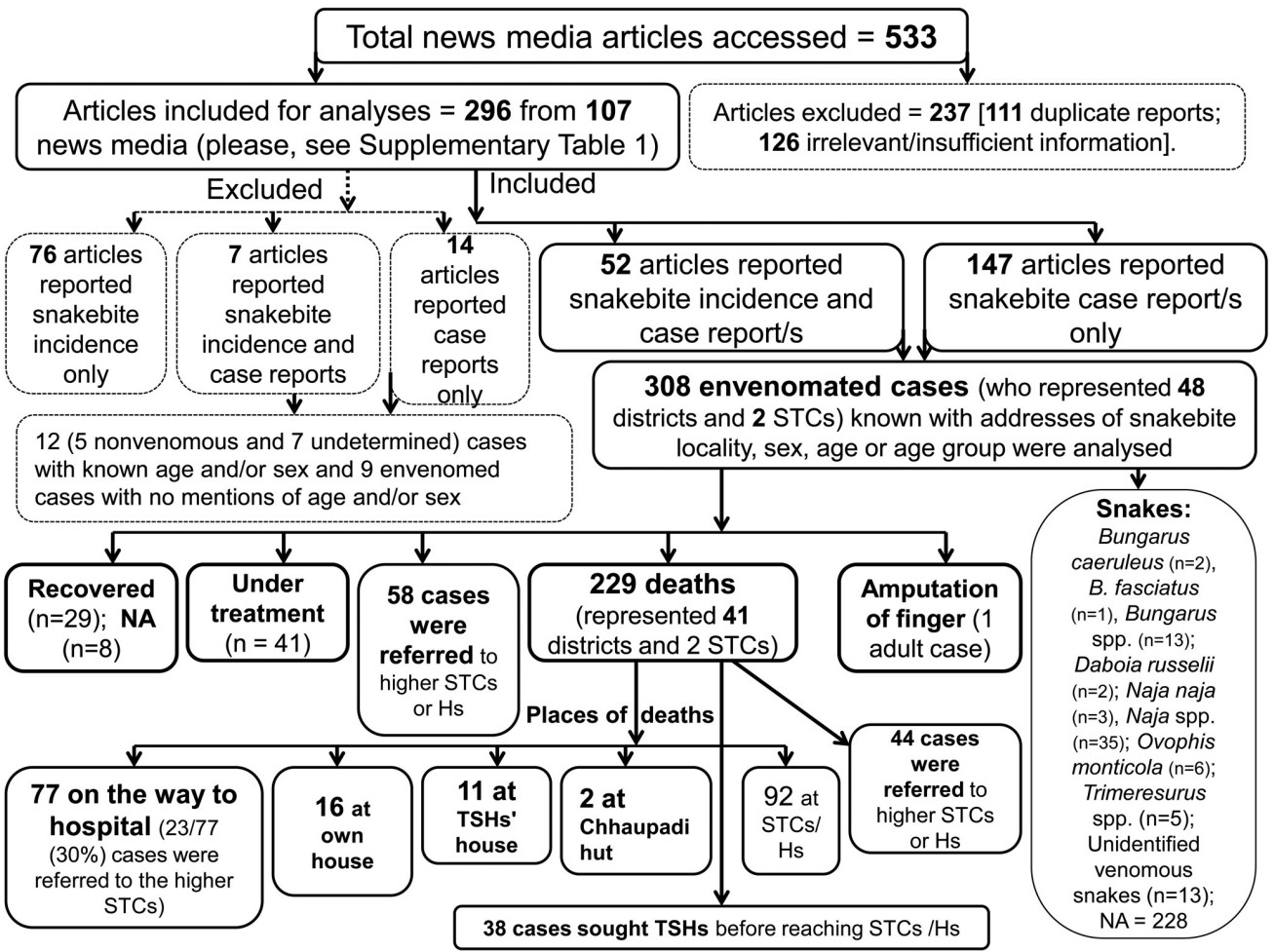

**Fig 2. Flow diagram showing eligible news media articles, snakebite incidences, case reports, and outcomes of envenomated cases with substantial information including snake species involved in envenoming [STCs: Snakebite Treatment Centers, Hs: hospitals where antivenom supply was unclear but snakebite cases accessed, NPOs: Nepal Police Offices; TSHs: traditional snakebite healers].** The details information used in this figures are available in Table 1 and S1–S3 Tables.

activeness and bitten body parts, activeness and time of snakebites, human habitations and time of envenomings, etc.) and the Pearson's Chi-square test for independence (PCTI) when the expected frequency of observations in each cells was at least 5 (variables: human habitations and sex, activeness of victims and age groups, activeness and localities of snakebites, etc.).

We performed all analyses with the R statistical program (R version 4.1.2 (2021-11-01), The R Foundation for Statistical Computing Platform), the Microsoft Excel, and the Google Earth Pro. All statistical tests were performed at 5% significance level. We round figured the p-values less than 0.001 to <0.001.

### Ethics statement

No ethical approval was sought because we analyzed publicly available data. We used name of patients and known geographic locations to cross-check and avoid duplications. The known geographic locations where the envenomings occurred were used to generate the approximate coordinates of localities and identify any urban areas where envenomings were commonly

reported. To support designing more sophisticated survey including snakebite affected urban areas, too, we mentioned some locality information up to the level of towns. However, considering ethics, we analyzed data anonymously and without mentioning the last level of administrative units.

## Results

We identified 199 news articles which described 308 envenomings with substantial information on demographics, circumstances, prehospital interventions, and outcomes (Fig 2). These envenomings were known to occur in 48 districts (Table 1) across the altitudinal ranges from 63–2870 m asl [median: 177, CI: 188–330, IQR: 98–506, Fig 3]. Snakebite envenomings more frequently occurred in tropical region [60–999 above sea level (asl); n = 263, 85%] followed by subtropical (1,000–1,999 asl; n = 35, 11%), and temperate regions (2,000–2,999 asl; n = 10, 3%) [p-value: < 0.001 (PCTGF)]. More than 10 envenomings and 10 deaths were reported from each of the seven districts (Table 1).

### i) Demographics

The median age of 249 patients was 19 y [CI: 21.5–27.5, range: 0.75–83 y, IQR: 10–40 y, Fig 4]. Among 59 cases without known age, 23 cases were with known age group (20/23 cases were known with sex; 3/23 children were reported without sex) and 36 cases were without known age group but with known sex in addition to other substantial information. Among a total of 249 cases with known age, the most victims were children (n = 115, 46%) with a median age of 9 y ranging from 0.75–17 y (CI: 7.8–9.5 y, IQR: 4–13 y), followed by adults [n = 74, 30%; median: 27.5 y, CI: 26.5–29 y, range: 18–40 y, IQR: 23–32 y], and elders [n = 60, 24%, median: 55 y, CI: 53.5–59.9 y, range: 41–83 y, IQR: 49.8–65 y]. Overall, children were mostly affected by snakebite envenomings in Nepal (p-value = <0.001, Table 2). However, the distribution of envenomings according to age groups was independent of both activeness of the victims (Table 3) and habitations used by them while snakebites (Table 4).

We determined that sex was known in 304 cases. Among four cases without sex, one case was with known age and three with age group only. Although slightly more bite victims were females (n = 160, 53%) than males, snakebite envenomings in males and females were not significantly different (p-value: 0.359, Table 2). Among a total of 248 cases with known sex and age, more victims were females (n = 133, 54%, with a median age of 18 y, CI: 18.5–25.5 y, IQR: 9–35 y, range: 0.75–78 y than males (n = 115, 46%, median: 23 y, CI: 23–33 y, range: 1–83 y; IQR: 11–50 y]. However, there was no association between sex and age-class of respective patients [p-value: 0.076 (PCTI)]. Further, the occurrence of envenomings according to females and males was independent of the activeness of victims (p-value: 0.591, Table 3) but dependent with the habitations used by victims while snakebites (p-value: 0.019, Table 4). Among 195 cases in which sex of envenomed patients and types of human habitations where snakebites occurred were known, envenomings were most often to the females at residential areas (n = 84, 43%, Table 4). Among known occupations, although students (n = 11, 44%) and farmers and workers (n = 6, 24%) were more frequently envenomed than other occupational people, the envenomings did not occur according to occupation (p-value: 0.089, Table 2). Occupational occurrences of envenomings were independent of victims' activeness (p-value: 0.381, Table 3) and use of places in and out of human habitations (p-value: 0.922, Table 4).

### ii) Circumstances of snakebite envenomings

   **a) Locality of snakebite envenomings.**   Fifty-five percent (n = 160) of envenomings occurred in urban and semi-urban areas (p-value: 0.068; we excluded 19 out of 308 (6.2%)

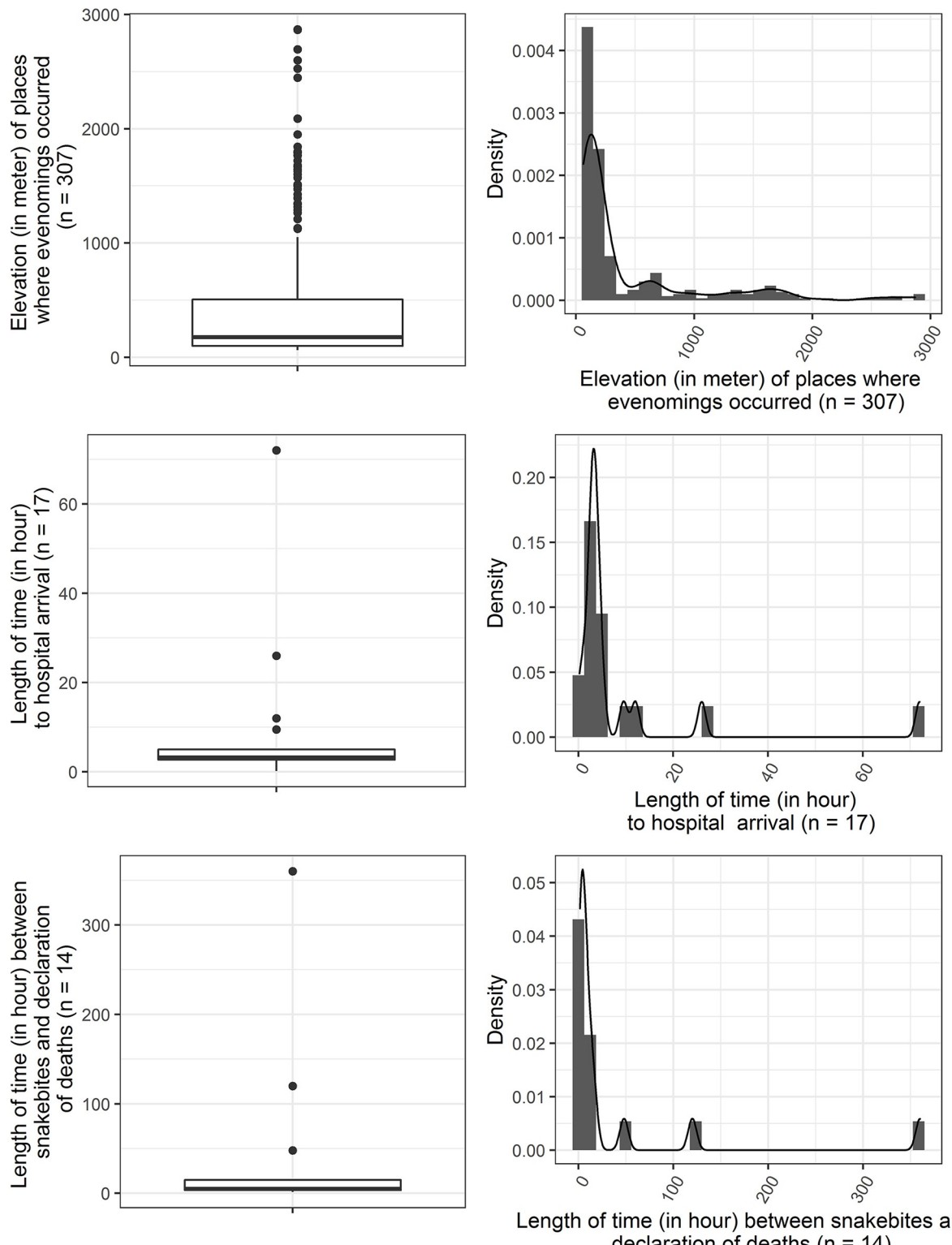

**Fig 3. Showing distribution of elevations (where snakebite envenomings occurred) and length of time between snakebite and hospital arrival and between snakebite and death.**

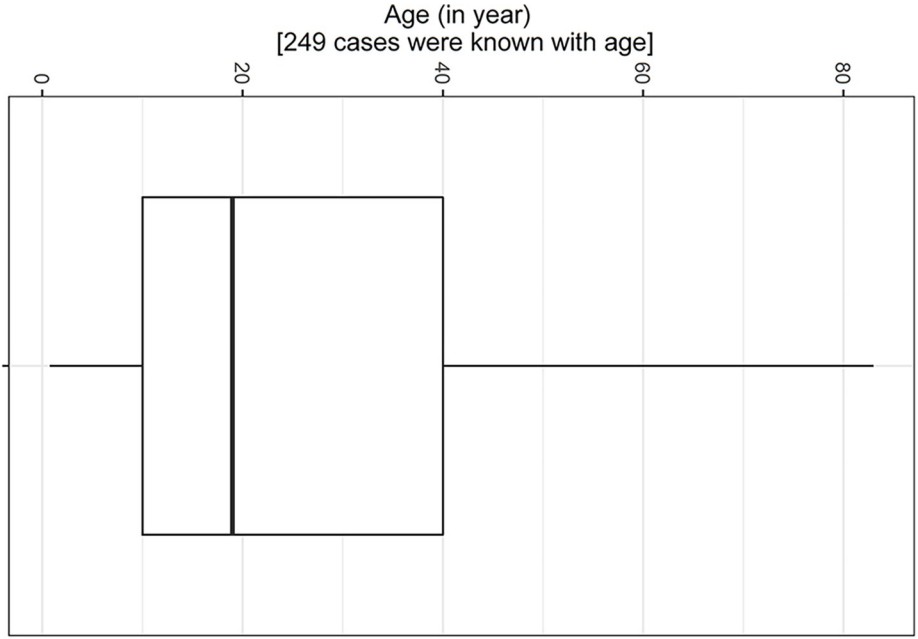

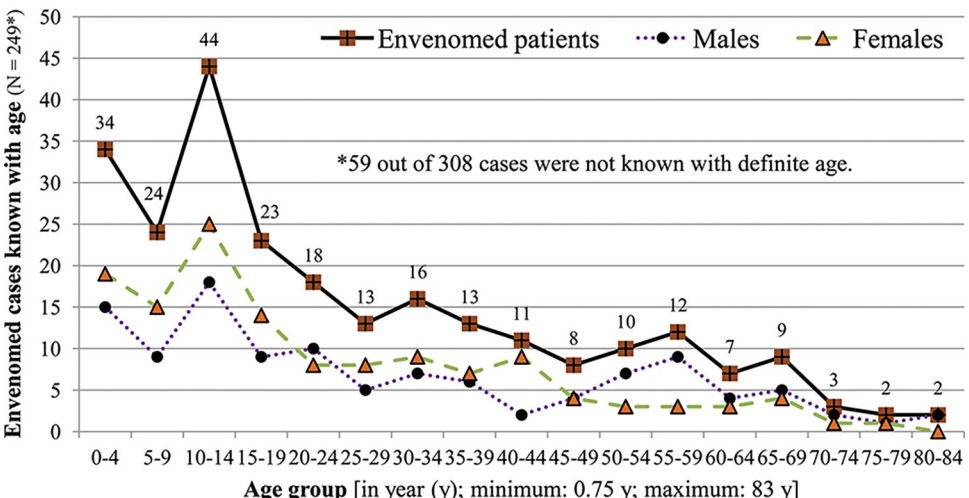

**Fig 4. Showing distribution of envenomated patients' age using box and whisker plot (above) and line curves (below).**

undefined localities [i.e., localities known with the district's name only (n = 18, 5.8%) and locality defined as Terai (n = 1, 0.3%)] while Chi-square tests (Table 2). We found four snake-bite deaths within a single day in Bardibas (n = 3) and Gaushala Municipalities (n = 1) of Mahottari District and three snakebite deaths in Garuda (n = 1) and Gadhimai Municipalities (n = 2) of Rautahat District in July 2018. In a similar outbreak, there were three snakebite deaths within a week from ward number one of Bardibas Municipality in Mahottari in August 2017 and 11 envenomings in a single day from Gaur Municipality in Rautahat in July 2015. In September 2020, a single death and two envenomings occurred in the same household on the same day in Sardanagar town of Bharatpur Metropolitan City in Chitwan District.

**Table 2. Demographics and circumstances of 308 envenomated cases.**

| Demographic and circumstantial features | | Number (%) | p-value (¥) |
|---|---|---|---|
| **A. Demographics** | | | |
| ***1. Age-classes*** [NM: 36] | | | |
| | Children (70 females, 52 males, 4 with undefined sex) | 126 (46.3) | <**0.001** |
| | Adults (49 females, 36 males) | 85 (31.3) | |
| | Elders (25 females, 36 males) | 61 (22.4) | |
| | Total | 272 (100) | |
| ***2. Sex*** [NM:4] | | | |
| | Females | 160 (52.6) | 0.359 |
| | Males | 144 (47.4) | |
| | Total | 304 (100) | |
| ***3. Occupation*** [NM: 283] | | | |
| | Students | 11 (44.0) | 0.089 |
| | Farmers and workers• | 6 (24.0) | |
| | Employed persons•• | 6 (24.0) | |
| | Others••• | 2 (8.0) | |
| | Total | 25 (100) | |
| **B. Circumstances of snakebites** | | | |
| ***1. Remoteness of snakebite localities*** | | | |
| | Urban and semi-urban areas (i.e., municipalities, sub-metropolitan and metropolitan cities) | 160 (55.4) | 0.068 |
| | Rural areas (village councils, aka rural municipalities) | 129 (44.6) | |
| | Total | 289 (100) | |
| ***2. Places*** [NM: 115] | | | |
| | In human inhabited areas | 141 (73.1) | <**0.001** |
| | Out of human inhabited areas | 52 (26.9) | |
| | Total | 193 (100) | |
| ***3. Activities*** [NM: 129] | | | |
| | Active status of victims while snakebite | 88 (49.2) | **0.823** |
| | Passive status of victims while snakebite | 91 (50.8) | |
| | Total | 179 (100) | |
| ***4. Bitten parts*** [NM: 261] | | | |
| | Upper extremities [hand (n = 19), finger (n = 5)] | 24 (51.1) | **0.017** |
| | Lower extremities [foot (n = 1), leg (n = 14)] | 15 (31.9) | |
| | Head and neck [earlobe (n = 2), cheek (n = 3), forehead (n = 1), neck (n = 2)] | 8 (17.0) | |
| | Total | 47 (100) | |
| ***5. Time of snakebite#*** [NM: 133] | | | |

(*Continued*)

**Table 2.** (Continued)

| Demographic and circumstantial features | | Number (%) | p-value (¥) |
|---|---|---|---|
| **A. Demographics** | | | |
| | Night (20:00–02:59 h) | 94 (53.7) | <**0.001** |
| | Day (10:00–16:59 h) | 36 (20.6) | |
| | Morning (05:00–09:59 h) | 25 (14.3) | |
| | Evening (17:00–19:59 h) | 15 (8.6) | |
| | Early morning (03:00–04:59 h) | 5 (2.9) | |
| | Total | 175 (100) | |
| *6. Months of snakebite* | | | |
| | July | 105 (34.1) | <**0.001** |
| | August | 60 (19.5) | |
| | June | 59 (19.2) | |
| | September | 43 (14.0) | |
| | May | 15 (4.9) | |
| | October | 15 (4.9) | |
| | November | 4 (1.3) | |
| | April | 3 (1.0) | |
| | March | 1 (0.3) | |
| | December | 1 (0.3) | |
| | February | 1 (0.3) | |
| | January | 1 (0.3) | |
| | Total | 308 (100) | |
| *7. Seasons of snakebite* | | | |
| | Monsoon (June–September) | 267 (86.7) | <**0.001** |
| | Winter (October–February) | 22 (7.1) | |
| | Pre-monsoon (March–May) | 19 (6.2) | |
| | Total | 308 (100) | |
| *8. Types of snakes* [^ = family] [NM: 228] | | | |
| *i) Elapidae^* | Cobras (*Naja* spp.) | 38 (56.7) | <**0.001** |
| | Kraits (*Bungarus* spp.) | 16 (23.9) | |
| *ii) Viperidae^* | Pitvipers [*Ovophis monticola* (Mountain Pitvipers, n = 6, 9.0%), *Trimeresurus* spp. (Green Pitvipers, n = 5, 7.5%)] | 11 (16.5) | |
| | True viper [*Daboia russelii* (Russell's Vipers)] | 2 (3.0) | |
| | Total | 67 (100) | |

**Abbreviations and symbols**:

**NM:** not mentioned (i.e., not available, NA)

• workers included 1 labor-worker and 1 worker known without defined type of work

•• the employment included 2 healthcare professionals (i.e., 1 midwife and 1 healthcare volunteer), 2 armed forces (i.e., 1 army and 1 police), 2 teachers

••• others included 1 social worker and 1 traditional healer

% = percent

¥ = Pearson's Chi-squared test for goodness of fit (PCTGF).

**b) Places and victims' activities when bitten.** The prominent risk places for snakebite envenoming in Nepal were human-inhabited areas (73%, p-value: <0.001, Table 2). The human-inhabited areas included indoor locations [n = 77, 39.9%: 49 (25.4%) indoor sleeping-beds and 27 (14%) non-specified indoor sites, and a hut in brick chimney (0.5%) where a worker was envenomed], houses and public buildings [n = 35, 18.1%: 33 (17.1%) houses without mention of specific locations, two (1%) public buildings (quarantine and hospital)], immediate vicinity of houses [n = 18, 9.3%: yard (n = 12, 6.2%), courtyard (n = 1, 0.5%), and the outskirts of house (n = 5, 2.6%)], cattle sheds and their vicinity [n = 7 (3.6%): cattle sheds (n = 6, 3.1%), and a place nearby pen i.e., enclosed space in which animals are kept (0.5%)], and chhaupadi hut (i.e., an isolated house being used by menstruating women/girls; n = 4, 2.1%]. About one-third (n = 52, 26.9%) of envenomings occurred in areas far from human inhabitation. These included 29 (15%) agricultural lands such as paddy, maize, watermelon, cardamom, and banana fields and vegetable farms, 11 (5.7%) nearby houses (but how far the snakebite sites from those houses were not indicated), 6 (3.1%) toilets [without specifying indoor/outdoor toilet or open toilet (n = 3, 1.6%) and open toilet in the field (n = 3, 1.6%), 4 (2.1%) on the road, and 2 (1%) outdoors (with no mention of specific areas). Snakebite envenomings in and out of human inhabitations were dependent on activeness of victims while snakebite (p-value: <0.001, Table 3). Residential areas were prone to envenomings while passive status of victims.

However, venomous snakebites may occur during both active and passive status of victims (p-value: 0.823, Table 2). The passive conditions of patients (n = 91, 50.8%) were sleeping indoor and on the outskirts (n = 81, 45.3%), resting at outskirts and quarantine (n = 8, 4.5%), watching television (n = 1, 0.6%), and performing security guard (n = 1, 0.6%) where as the active status of the patients while snakebites (n = 88, 49.2%) included agriculture and labor-worker related activities (n = 34, 19%) such as agricultural activities (n = 26, 14.5%), caring cattle (n = 1, 0.6%), cooking cattle fodder (n = 1, 0.6%), fetching pile of straw to fed cattle (n = 1, 0.6%), filling up a small mammalian hole (n = 1, 0.6%), giving fodders to cattle (n = 1, 0.6%), shifting herd of goat (n = 1, 0.6%), working in cow shed (n = 1, 0.6%), and labor-workers' activities (n = 1, 0.6%), household activities (n = 14, 7.8%) that included household chores (n = 10, 5.6%), pulling a mattress while dinner time (n = 1, 0.6), pulling cover of indoor rice-store (n = 1, 0.6), throwing dust (n = 1, 0.6), and sweeping (n = 1, 0.6), playing on the yard, nearby home, and outdoor (n = 19, 10.6%), walking (n = 6, 3.4%): on the road (n = 5, 2.8%), and walking to the cattle shed (n = 1, 0.6%), going or using toilet (n = 6, 3.4%), and disturbing snakes knowingly or unknowingly (n = 9, 5%): treading snake (n = 3, 1.7%), inserting hand in a small mammalian hole (n = 2, 1.1%), rescuing snakes (n = 2, 1.1%), putting leg in the hole of rodents (n = 1, 0.6%), and warding off the intruded elephant (n = 1, 0.6%). Except two envenomings that occurred during the victims' intentional interaction with snakes, all of these bites occurred during natural human activities. A farmer bitten by a Common Cobra (*N. naja*) in paddy field during his agricultural activities bit the cobra repeatedly and intentionally until the responsible snake was dead by following the advice of a snake charmer.

## c) Bitten body parts

Among 47 cases in which body parts involved and the activeness of victims while snakebites were known, bites were most often to the upper extremities (p-value: 0.017, Table 2). Which body parts were bitten by snake was dependent on activeness of victims (p-value: 0.017, Table 3). Among inactive victims, the head and neck were bitten whereas extremities were often bitten while victims were active. But, there was no association between involvement of body parts and types of habitations used by victims while snakebites (p-value = 0.378, Table 4).

**Table 3. Association of demographics and circumstances of snakebite envenomings with activeness of victims.**

| Study variables | Activeness of victims | | Total number (%) | p-value¥ |
|---|---|---|---|---|
| | Active [n(%)] | Passive [n(%)] | | |
| **A. Demographics** | | | | |
| *1. Age-classes* | | | | |
| Children | 33 (20.9) | 36 (22.8) | 69 (43.7) | 0.434 (PCTI) |
| Adults | 21 (13.3) | 31 (19.6) | 52 (32.9) | |
| Elders | 20 (12.7) | 17 (10.8) | 37 (23.4) | |
| Total | 74 (46.8) | 84 (53.2) | 157 (100) | |
| *2. Sex* | | | | |
| Females | 49 (27.4) | 46 (25.7) | 95 (53.1) | 0.591 (PCTI) |
| Males | 39 (21.8) | 45 (25.1) | 84 (46.9) | |
| Total | 88 (49.2) | 91 (50.8) | 179 (100) | |
| *3. Occupation* | | | | |
| Students | 5 (23.8) | 4 (19.0) | 9 (42.9) | |
| Farmers and workers• | 4 (19.0) | 2 (9.5) | 6 (28.6) | 0.381 (FET) |
| Employed persons•• | 1 (4.8) | 4 (19.0) | 5 (23.8) | |
| Others••• | 1 (4.8) | 0 (0.0) | 1 (4.8) | |
| Total | 11 (52.4) | 10 (47.6) | 21 (100) | |
| **B. Circumstances of snakebites** | | | | |
| *1. Remoteness of snakebite localities* | | | | |
| Urban and semi-urban areas‖ | 43 (24.7) | 54 (31.0) | 97 (55.7) | 0.236 (PCTI) |
| Rural areas† | 42 (24.1) | 35 (20.1) | 77 (43.3) | |
| Total | 85 (48.9) | 89 (51.1) | 174 (100) | |
| *2. Localities* | | | | |
| In human inhabited areas | 40 (22.3) | 90 (50.2) | 130 (72.5) | <**0.001** (PCTI) |
| Out of human inhabited areas | 48 (26.8) | 1 (0.6) | 49 (27.5) | |
| Total | 88 (49.2) | 91 (50.8) | 179 (100) | |
| *3. Bitten parts* | | | | |
| Upper extremities | 13 (30.2) | 8 (18.6) | 21 (48.8) | **0.017** (FET) |
| Lower extremities | 12 (27.9) | 2 (4.7) | 14 (32.6) | |
| Head and neck | 2 (4.7) | 6 (13.9) | 8 (18.6) | |
| Total | 27 (62.8) | 16 (37.2) | 43 (100) | |
| *4. Time of snakebite* | | | | |
| Night | 10 (6.6) | 74 (49.0) | 84 (55.6) | <**0.001** (FET) |
| Day | 28 (18.5) | 2 (1.3) | 30 (29.9) | |
| Morning | 18 (11.9) | 3 (2.0) | 21 (13.9) | |
| Evening | 10 (6.6) | 1 (0.7) | 11 (7.3) | |
| Early morning | 1 (0.7) | 4 (2.6) | 5 (3.3) | |
| Total | 67 (44.4) | 84 (55.6) | 151 (100) | |
| *5. Months of snakebite* (Absent months below had no data for month of snakebite and activeness of the patients) | | | | |
| April | 0 (0.0) | 1 (0.6) | 1 (0.6) | 0.290 (FET) |
| August | 18 (10.1) | 16 (8.9) | 34 (19.0) | |
| July | 33 (18.4) | 31 (17.3) | 64 (35.8) | |
| June | 21 (11.7) | 15 (8.4) | 36 (20.1) | |
| May | 5 (2.8) | 5 (2.8) | 10 (5.6) | |
| October | 4 (2.2) | 5 (2.8) | 9 (5.0) | |
| September | 7 (3.9) | 18 (10.1) | 25 (14.0) | |
| Total | 88 (49.2) | 91 (50.8) | 179 (100) | |

*(Continued)*

**Table 3.** (Continued)

| Study variables | Activeness of victims | | Total number (%) | p-value¥ |
|---|---|---|---|---|
| | Active [n(%)] | Passive [n(%)] | | |
| *6. Seasons of snakebite* | | | | |
| Monsoon | 79 (44.1) | 80 (44.7) | 159 (88.8) | 0.924 (PCTI) |
| Winter | 5 (2.8) | 6 (3.4) | 11 (6.1) | |
| Pre-monsoon | 4 (2.2) | 5 (2.8) | 9 (5.1) | |
| Total | 88 (49.1) | 91 (50.9) | 179 (100) | |
| *7. Snake types* | | | | |
| Cobras | 19 (36.5) | 8 (15.4) | 27 (51.9) | 0.102 (FET) |
| Kraits | 5 (9.6) | 8 (15.4) | 13 (25.0) | |
| Pitvipers | 8 (15.4) | 2 (3.8) | 10 (19.2) | |
| True vipers | 2 (3.8) | 0 (0.0) | 2 (3.8) | |
| Total | 34 (65.4) | 18 (34.6) | 52 (100) | |

**Abbreviations and symbols**:

% = percent

¥ = Pearson's Chi-squared test for goodness of fit

• workers included 1 labor-worker and 1 worker known without defined type of work

•• the employment included 2 healthcare professionals (i.e., 1 midwife and 1 healthcare volunteer), 2 armed forces (i.e., 1 army and 1 police), 2 teachers

••• others included 1 social worker and 1 traditional healer

‖ = municipalities and metropolitan cities

† = village councils, aka rural municipalities.

### d) Temporal patterns of snakebite envenomings

A total of 267 out of 308 (87%) envenomings occurred during rainy months of the year, mainly during the night (p-value: <0.001 each, Table 2). Although envenomings occurred in all periods of the day throughout the year, the peak of envenomings occurred during July, August, and June and at night (Table 2). Envenomings that occurred according to time periods were dependent on activeness and types of habitations used by victims (p-value: <0.001 each, Tables 3 and 4). These periods, however, were not associated with the months [p-value: 0.608 (FET)] and seasons [p-value = 0.669 (FET] of snakebite envenomings.

### e) Snakes responsible for envenomings

Sixty-seven out of 308 (22%) victims were envenomated by known types of snakes. Among 67 venomous snakes, the majority of snakes belonged to elapidae family [n = 54 (80.6%): cobras (*Naja* spp., n = 38, 56.7%); kraits (*Bungarus* spp., n = 16, 23.9%)], followed by snakes belonged to viperidae family: (n = 11, 19.5%, p-value = <0.001, Table 2). Among the viperids, pitvipers [i.e., Mountain Pitviper (*Ovophis monticola*, n = 6, 9%) and Green Pitvipers (*Trimeresurus* spp., n = 5, 7.5%)] were commonly involved in envenomings. Only two (3%) individuals of true vipers [i.e., Russell's Vipers (*Daboia russelii*)] caused envenomings. Although envenomings often occurred due to cobras and kraits (p-value = <0.001, Table 2), activeness (p-value: 0.102, Table 3) and areas used by victims while snakebite (0.324, Table 4) were independent of involvement of venomous snakes' types in bites. Thirteen cases were reported to be bitten by unidentified venomous snakes. Types of snakes were not mentioned in reports of 228 envenomed cases. There were no reports of rear-fanged snakes although they are present in Nepal but apparently not envenoming people.

**Table 4. Relatedness of demographics and circumstances of snakebites with habitations used by envenomated victims.**

| Study variables | Habitations used by victims while snakebite envenomings | | Total number (%) | p-value¥ |
|---|---|---|---|---|
| | In [n(%)]Ω | Out [n(%)]Ω | | |
| **A. Demographics** | | | | |
| *1. Age-classes* | | | | |
| Children | 59 (33.9) | 17 (9.8) | 76 (43.7) | 0.983 (PCTI) |
| Adults | 43 (24.7) | 13 (7.5) | 56 (32.2) | |
| Elders | 32 (18.4) | 10 (5.7) | 42 (24.1) | |
| Total | 134 (77) | 40 (23.0) | 174 (100) | |
| *2. Sex* | | | | |
| Females | 84 (43.1) | 20 (10.3) | 104 (53.3) | **0.019** (PCTI) |
| Males | 59 (30.3) | 32 (16.4) | 91 (46.7) | |
| Total | 143 (73.3) | 52 (26.7) | 195 (100) | |
| *3. Occupation* | | | | |
| Students | 7 (29.2) | 3 (12.5) | 10 (41.7) | 0.922 (FET) |
| Farmers and workers• | 5 (20.8) | 1 (4.2) | 6 (25.0) | |
| Employed persons•• | 4 (16.7) | 2 (8.3) | 6 (25.0) | |
| Others••• | 1 (4.2) | 1 (4.2) | 2 (8.3) | |
| Total | 17 (70.8) | 7 (29.2) | 24 (100) | |
| **B. Circumstances of snakebites** | | | | |
| *1. Remoteness of snakebite localities* | | | | |
| Urban and semi-urban areas‖ | 78 (41.3) | 26 (13.7) | 104 (55.0) | 0.877 (PCTI) |
| Rural areas† | 62 (32.8) | 23 (12.2) | 85 (45.0) | |
| Total | 140 (74.1) | 49 (25.9) | 189 (100) | |
| *2. Activeness of victims* | | | | |
| Active | 40 (22.3) | 48 (26.8) | 88 (49.2) | **<0.001** (PCTI) |
| Passive | 90 (50.3) | 1 (0.6) | 91 (50.8) | |
| Total | 130 (72.6) | 49 (27.4) | 179 (100) | |
| *3. Bitten parts* | | | | |
| Upper extremities | 14 (31.8) | 8 (18.2) | 22 (50.0) | 0.378 (FET) |
| Lower extremities | 8 (18.2) | 6 (13.6) | 14 (31.8) | |
| Head and neck | 7 (15.9) | 1 (2.3) | 8 (18.2) | |
| Total | 29 (65.9) | 15 (34.1) | 44 (100) | |
| *4. Time of snakebite* | | | | |
| Night | 84 (52.8) | 5 (3.1) | 89 (56.0) | **<0.001** (FET) |
| Day | 8 (5.0) | 23 (14.5) | 31 (19.5) | |
| Morning | 16 (10.1) | 6 (3.8) | 22 (13.8) | |
| Evening | 9 (5.7) | 3 (1.9) | 12 (7.5) | |
| Early morning | 4 (2.5) | 1 (0.6) | 5 (3.1) | |
| Total | 121 (76.1) | 38 (23.9) | 159 (100) | |
| *5. Months of snakebite* (Absent months below had no data for snakebites and activeness of the patients) | | | | |
| April | 1 (0.5) | 0 | 1 (0.5) | 0.620 (FET) |
| August | 25 (12.8) | 13 (6.7) | 38 (19.5) | |
| July | 48 (24.6) | 19 (9.7) | 67 (34.4) | |
| June | 28 (14.4) | 12 (6.2) | 40 (20.5) | |
| May | 9 (4.6) | 2 (1.0) | 11 (5.6) | |
| October | 8 (4.1) | 2 (1.0) | 10 (5.1) | |
| September | 24 (12.3) | 4 (2.1) | 28 (14.4) | |

*(Continued)*

**Table 4.** (Continued)

| Study variables | Habitations used by victims while snakebite envenomings | | Total number (%) | p-value¥ |
|---|---|---|---|---|
| | In [n(%)]Ω | Out [n(%)]Ω | | |
| Total | 143 (73.3) | 52 (26.7) | 195 (100) | |
| **6. Seasons of snakebite** | | | | |
| Monsoon | 125 (64.1) | 48 (24.6) | 173 (88.7) | 0.684 (FET) |
| Winter | 8 (4.1) | 2 (1.0) | 10 (5.1) | |
| Pre-monsoon | 10 (5.1) | 2 (1.0) | 12 (6.2) | |
| Total | 143 (73.3) | 52 (26.7) | 195 (100) | |
| **7. Snake types** | | | | |
| Cobras | 21 (37.5) | 9 (16.1) | 30 (53.6) | 0.324 (FET) |
| Kraits | 10 (17.9) | 3 (5.3) | 13 (23.2) | |
| Pitvipers | 5 (8.9) | 6 (10.7) | 11 (19.6) | |
| True vipers | 2 (3.6) | 0 (0.0) | 2 (3.6) | |
| Total | 38 (67.9) | 21 (32.1) | 56 (100) | |

**Abbreviations and symbols**

Ω = **In** for human residential areas such as yard, indoor, etc. and **Out** for out of human residential areas such as crop fields, roads, etc.

% = percent

¥ = Pearson's Chi-squared test for goodness of fit

• workers included 1 labor-worker and 1 worker known without defined type of work

•• the employment included 2 healthcare professionals (i.e., 1 midwife and 1 healthcare volunteer), 2 armed forces (i.e., 1 army and 1 police), 2 teachers

••• others included 1 social worker and 1 traditional healer

‖ = municipalities and metropolitan cities

† = village councils, aka rural municipalities

**iii) Time to hospital arrival (DOH) and treatment seeking behavior.** The median DOH known for 17 cases was 3.3 h (CI: 2.8–13.50 h, range: 0.2–72 h, IQR: 2.8–5 h, NA = 291, Fig 4). Among 89 envenomed cases seeking traditional snakebite healers (TSHs) before reaching STCs or healthcare institution with no specialty, 51 (57%) cases consulted TSHs, whereas 38 (43%) cases accessed STCs without consultation of TSHs. No information regarding the consultation about TSHs was given for remaining 219 cases (Table 5). Altogether, 256 out of 284 (90%) envenomed cases sought treatment at a or healthcare institution with no specialty (Table 5, Fig 1). Thirty-four of 256 (13%) cases also sought TSHs before reaching healthcare institutions. Twenty-eight out of 284 (10%) cases did not seek modern care for snakebites. Among these 28 cases, 17 (61%) patients followed up TSHs (Information regarding treatment-seeking behavior was not mentioned for remaining 11 cases). For remaining 24 cases, information about modern snakebite treatment-seeking behavior was not available (Table 5). The modern treatment seeking of these victims was dependent on their consultation with TSHs (p-value = 0.001, PCTI).

**iv) Outcomes.** Table 5 illustrates the outcomes of envenomated cases with substantial information. Fifty-eight out of 143 (41%) snakebite cases were referred by STCs or hospitals (where patients were initially carried) to higher healthcare centers for snakebite treatments. Eight of those 58 (14%) cases being referred to higher healthcare centers visited TSHs first. The outcomes were dependent on the referral of the cases (p-value: 0.005, Table 5). Forty-four out of 58 (76%) referred cases died (23 cases died on the way to higher healthcare centers, 20 cases died in healthcare institutions during treatment, and the place of death was not mentioned in a case).

Among 198 envenomed cases with known places of deaths (Fig 2), deaths often occurred in STCs/hospitals (n = 92, 46.5%) followed by on the way to STC/hospitals (n = 77, 38.9%), own house (n = 16, 8.1%), traditional healers' house (n = 11, 5.6%), and chhaupadi hut (n = 2, 1.0%) (p-value: <0.001, PCTGF). Among a total of 77 deaths on the way to STCs/hospitals, 13 (17%) cases sought traditional healing prior to seeking antivenom therapy for snakebite envenomings.

Among 89 cases with known information on seeking TSHs before reaching STC/Hs, 51 cases (57%) consulted TSHs. 38 out of 51 (75%) cases who consulted TSHs died (p-value: 0.375, Table 5). Among 51 cases consulting TSH (Table 5), 17 (33%) cases visited TSHs only (i.e., they were not carried to STCs/hospitals) and had a 100% fatality rate. Remaining 34 (67%) cases who visited THS and then carried to STCs/hospitals had 62% (n = 21) fatality and 12% recovery (n = 4) [five cases were under treatment, and outcomes were not mentioned in four cases]. All 38 cases who did not consult TSH (Table 5) and directly visited STCs/hospitals had 66% (n = 25) fatality and 11% (n = 4) recovery [7 cases were under treatment, and outcomes were not mentioned in one case, and one patient's finger was amputated]. All 28 cases who did not seek modern care for snakebites died (p-value: 0.002, Table 5).

Among deaths, children and females had the highest fatality (p-value: 0.002, Table 5). These fatalities were more confined in tropical region [n = 198 (64%)] than in subtropical [n = 26 (8%)] and temperate regions [n = 5 (2%)] of this country [p-value: <0.001 (PCTGF)]. These fatalities were dependent on remoteness of the snakebites (p-value: 0.045, Table 5) and activeness of the victims (p-value: 0.043, Table 5). The most of deaths occurred among victims who were passive (mostly sleeping) while snakebite. Further, among the recovered cases, active cases were mostly recovered compared to the passive victims.

The greatest number of deaths occurred during the rainy season [n = 203 (68%)] followed by pre-monsoon [n = 14 (5%)], and winter [n = 12 (4%)] [p-value: 0.042, Table 5]. Of the 46 fatal bites, 27 (45%) were attributed to the cobra (*Naja* spp. and 13 (22%) to the kraits (*Bungarus* spp., Table 5). The median LOD was 6 h which was known for 14 patients (CI: 4.3–62.5 h, IQR: 4.1–15.5 h, range: 1.5–360 h, NA = 294, Fig 4). Among 300 cases known with outcomes of envenomings, 29 (10%) cases were recovered and 41 (14%) were under treatment (Table 5). One of the recovered cases was discharged with amputation of a finger.

## Discussion

This is the first study to analyze news media-reported Nepalese snakebite envenomings with substantial information. Dependency on traditional snakebite healers, the long distance between snakebite localities and STCs, and snakebite occurred at night were the major barriers to accessing healthcare facilities by people bitten by snakes in Nepal. These barriers are also common in India and Bangladesh [41]. Hence, this study's findings have significant policy implications in Nepal and other countries where the socio-economic, cultural, and geo-climate context of envenoming and associated consequences are similar.

### i) Demographics

Highly productive population of Nepalese communities were largely affected by snakebite envenomings [95% CI: 21.5–27.5 y]. This has direct impact on national economy of this country. Our findings of more children than other age classes being affected by snakebite envenomings mostly in and around houses are analogous to carelessness of minors and snake preferred surroundings in and around houses of Nepal [10,21]. A multicluster survey carried out in Terai of Nepal [6,42] also reported children at risk of venomous snakebites in Nepal.

**Table 5. Outcomes of envenomated patients.**

| Factors affecting outcomes of envenomings | Outcomes of envenomings (n = 300; information on outcomes of 8 cases was not available) | | | | |
|---|---|---|---|---|---|
| | Deaths (n = 229, 76.3%) | Recovered cases (n = 29, 9.7%) | Under treatment (n = 41, 13.7%) | Total number (%) | p-value |
| **I. Interventions at healthcare facilities** | | | | | |
| *A. Referred to other hospital* | | | | | |
| Yes (n = 58, 40.6%) | 44 (32.1) | 3 (2.2) | 6 (4.4) | 53 (38.7) | **0.005** |
| No (n = 85, 59.4%) | 49 (35.8) | 22 (16.1) | 13 (9.5) | 84 (61.3) | (PCTI) |
| Total [143 (100%)] | 93 (67.9) | 25 (18.2) | 19 (13.9) | 137 (100) | |
| **II. Prehospital interventions** | | | | | |
| *A. Seeking traditional snakebite healers before reaching STC/Hs* | | | | | |
| Yes (n = 51, 57.3%) | 38 (45.2) | 4 (4.8) | 5 (6.0) | 47 (56.0) | 0.375 |
| No (n = 38, 42.7%) | 25 (29.8) | 5 (6.0) | 7 (8.3) | 37 (44.0) | (PCTI) |
| Total [89 (100%)] | 63 (75.0) | 9 (10.7) | 12 (14.3) | 84 (100) | |
| *B. Seeking modern care of snakebite at STCs/hospitals* | | | | | |
| Yes (n = 256, 90.1%) | 178 (64.3) | 30 (10.8) | 41 (14.8) | 250 (90.9) | **0.002** |
| No (n = 28, 9.9%) | 28 (10.1) | 0 (0) | 0 (0) | 28 (10.1) | (FET) |
| Total [284 (100%)] | 206 (74.4) | 30 (10.8) | 41 (14.8) | 277 (100) | |
| **III. Demographics** | | | | | |
| *A. Age-class* | | | | | |
| Children (n = 126, 46.3%) | 114 (43.2) | 4 (1.5) | 7 (2.7) | 125 (73.3) | **0.002** |
| Adults (n = 85, 31.3%) | 61 (23.1) | 5 (1.9) | 16 (6.1) | 82 (31.1) | (FET) |
| Elders (n = 61, 22.4%) | 41 (15.5) | 4 (1.5) | 12 (4.5) | 57 (21.6) | |
| Total [272 (100%)] | 216 (81.8) | 13 (4.9) | 35 (13.3) | 264 (100) | |
| *B. Sex* | | | | | |
| Females (n = 160, 52.6%) | 126 (42.6) | 9 (3.0) | 19 (6.4) | 154 (52.0) | **0.020** |
| Male (n = 144, 47.4%) | 99 (33.4) | 21 (7.1) | 22 (7.4) | 142 (48.0) | (PCTI) |
| Total [304 (100%)] | 225 (76.1) | 30 (10.1) | 41 (13.8) | 296 (100) | |
| **IV. Circumstances** | | | | | |
| *A. Remoteness of snakebite localities* | | | | | |
| Urban and semi-urban areas‖ [N = 160 (55.4)] | 129 (45.6) | 11 (3.9) | 18 (6.4) | 158 (55.8) | **0.045** |
| Rural areas† [n = 129 (44.6)] | 87 (30.7) | 18 (6.0) | 20 (7.1) | 125 (44.2) | (PCTI) |
| Total [289 (100%)] | 216 (76.3) | 29 (10.2) | 38 (13.5) | 283 (100) | |
| *B. Activeness of victims* | | | | | |
| Active (n = 88, 49.2%) | 63 (35.6) | 9 (5.1) | 14 (7.9) | 86 (48.6) | **0.043** |
| Passive (n = 91, 50.8%) | 78 (41.1) | 2 (1.1) | 11 (6.2) | 91 (51.4) | (PCTI) |
| Total [179 (100%)] | 141 (79.7) | 11 (6.2) | 25 (14.1) | 177 (100) | |
| *C. Seasons* | | | | | |
| Monsoon (June–September; n = 267, 87%) | 203 (67.6) | 21 (7.0) | 38 (12.7) | 262 (87.3) | **0.042** |
| Pre-monsoon (March–May; n = 19, 6%) | 14 (4.7) | 4 (1.3) | 1 (0.3) | 19 (6.3) | (FET) |
| Winter (October–February; n = 22, 7%) | 12 (4.0) | 5 (1.7) | 2 (0.7) | 19 (6.4) | |
| Total [308 (100%)] | 229 (76.3) | 30 (10.0) | 41 (13.7) | 300 (100) | |
| *D. Types of snakes* | | | | | |
| *i) Elapidae family* | *40 (66.7)* | *6 (10.0)* | *4 (6.7)* | | 0.228 |
| Cobras [*Naja* spp. (n = 38, 56.7%)] | 27 (45.0) | 5 (8.2) | 2 (3.3) | 34 (56.7) | (FET) |
| Kraits [*Bungarus* spp. (n = 16, 23.9%)] | 13 (21.7) | 1 (1.7) | 2 (3.3) | 16 (26.7) | |
| *ii) Viperidae family* | *6 (10.0)* | *2 (3.3)* | *2 (3.3)* | | |
| Pitvipers [*Ovophis monticola* (n = 6, 9.0%); *Trimeresurus* spp. (n = 5, 7.5%)] | 6 (10.0) | 1 (1.7) | 2 (3.3) | 9 (15.0) | |
| Russell's Viper [*Daboia russelii* (n = 2, 3.0%)] | 0 (0) | 1 (1.7) | 0 (0) | 1 (1.7) | |

*(Continued)*

**Table 5.** (Continued)

| Factors affecting outcomes of envenomings | Outcomes of envenomings (n = 300; information on outcomes of 8 cases was not available) | | | | |
|---|---|---|---|---|---|
| | Deaths (n = 229, 76.3%) | Recovered cases (n = 29, 9.7%) | Under treatment (n = 41, 13.7%) | Total number (%) | p-value |
| Total [67 (100%)] | 46 (76.7) | 8 (13.3) | 6 (10.0) | 60 | |

**Abbreviations and symbols**:

**Hs:** hospitals where antivenom supply was unclearly reported but snakebite cases accessed those health institutions

**NA:** not available (i.e., not mentioned)

**STCs:** Snakebite Treatment Centers

% = percent

‖ = municipalities and metropolitan cities

† = village councils, aka rural municipalities.

We found both sexes being equally vulnerable to snakebite envenomings (Table 2). However, there are reports of female dominance in recently conducted studies in different parts of Nepal [a survey carried out in the Terai [6,42] and a study of snakebite envenomings admitted in Bheri Hospital having service areas in western Terai to lower-middle hills of Nepal [11]]. Other distant past studies (a study of snakebite envenomings admitted in Bharatpur Hospital having service areas in south-central Terai to lower-middle hills of Nepal [43] and the subsequent study of confirmed Common Kraits and Russell's Viper envenoming known from western, central, and eastern Nepal [10]) showed male dominance. Interestingly, we found females being mostly affected by snakebite envenomings when these episodes occurred in residential areas (p-value: 0.019, Table 4). The prevalent patriarchal culture leading to an increase of female activities and trends of more out-migrating male laborers than female laborers [11,44] might reflect the slight female preponderance of snakebite envenomings in and immediate surroundings of houses of Nepal. This suggests that females engaged in residential areas are at risk of venomous snakebites reflecting influence of patriarchal communities in Nepal where most of works at residential areas are often allocated to females.

Although there were frequent reports of students and farmers being mostly envenomed in different parts of Nepal [A Ph.D. dissertation carried out in eastern, central, and some parts of western Nepal [10] and next study carried out in a referral hospital in the mid-western Nepal [11] reported envenomed students (27–35%) and farmers (24–43%)], we found all occupational people being equally vulnerable to venomous snakebites (Table 2). Our analyses showed that people engaged in non-agricultural occupations (Table 2) were also affected by snakebite envenoming. Journalists largely missed reporting occupations of the snakebite victims, although it was essential to understand the impact of snakebite in occupational groups. However, educational intervention for prehospital care and pragmatic prevention of snakebites in residential areas in particular should include children and consider socio-cultural contexts to minimize the risks of envenomings and associated deaths.

### ii) Circumstances of snakebite envenoming

We identified that the envenomings in rural and non-rural (i.e., urban and semi-urban) areas of Nepal are not significantly different (p-value: 0.068, Table 2). Further, remoteness of envenomings was independent of victims' activeness (p-value: 0.236, Table 3) and their use of in and out of human habitations (p-value: 0.877, Table 4). The metropolitan cities, sub-metropolitan cities, and municipalities of Nepal include both urban and semiurban areas (sometimes, extremely rural areas, too). Therefore, most of these administrative bodies located in tropical

Terai region have agrarian communities and possess greater diversity and abundance of medically relevant snakes [10,22]. A Ph.D. dissertation also reported several confirmed Common Krait and Russell's Viper bites from urban, semi-urban, and rural areas of Nepal [10]. Next, our analyses declared an 'outbreak of snakebite envenomings' in urban areas of three districts. By translating a clinical consensus from an international panel of toxicologists who recommended as few as three toxic cases within 72 hours appearing in the same city to consider an 'outbreak of toxicity' [45], we believe that there exist outbreaks of snakebite envenomings in several urban localities of Nepal due to intensive activities of snakes and humans, particularly during the monsoon at dark hours of the day. Alike our findings, Igawe et al. [46] reported snakebite outbreak in the 9th, 19th, and 25th of May, 2016 in communities (wards) of the Donga Local Government Area of Nigeria. Therefore, our findings highlight the severe snakebite aspects of envenomings and the need to empower healthcare systems urgently. Those outbreaks, even in urban areas, suggest that large-scale community-based surveys [6] should also include cities and towns because the urban periantrhopic ecosystems particularly in the high snakebite prone tropical Terai districts of Nepal might be associated with the human-venomous snake conflicts. But, a recently conducted multicluster random survey of snakebites in Terai of Nepal excluded urban communities [6]. Therefore, our findings suggest the need of designing a large-scale, nationally representative epidemiological survey including snakebite prone rural as well as non-rural areas of this country.

Our district-wise incidence of those envenomings (Table 1) can support Nepalese medical authorities to estimate antivenom and healthcare personnel requirements and their appropriate distribution in the snakebite affected areas because snakebite risk area is extended in some parts of Nepal (Fig 5). The recent habitat suitable modeling for medically relevant snakes of Iran [47] showed similar extension of snakebite risks in its mountains. The records of envenomings and deaths in the higher hills and mountain ranges of Nepal (Fig 5) suggest transcending of venomous snakes from Terai, probably due to the increment of road transport in the Chure and Mahabharat ranges of Nepal (Fig 1). Snakes could be transferred along with goods carried in trucks. Further, global warming might support their flourishing in those regions [48]. For the effective snakebite management in districts representing higher hills and mountains (Table 1), there is need of immediate response of Nepal Government authorities.

Human-inhabited areas particularly in tropical regions of Nepal were at the highest risk for snakebite envenomings. These envenomings mostly occurred while people sleep (Tables 2 and 3). Similar observations were reported in a hospital-based study of envenomated cases from 11 districts [11] representing far- and mid-western Nepal and a community survey in 23 districts [6] across the Terai region of Nepal. Although envenomings while victims' active and passive conditions were not significantly different (p-value: 0.823, Table 2), people should be wary at bed time in residential areas, particularly in the tropical regions of Nepal because we found the maximum envenoming while sleeping (p-value: <0.001, Table 3).

Unlike a report from a similar study in USA [17], we found rare intentional human-snake interactions causing envenomings. However, in western Nepal, some harmful religio-cultural practices, such as sleeping/resting in Chhaupadi hut (a special house used while menstruating by women and girls in the Chhaupadi culture) [49], and reverence for venomous snakes residing on the premises [21] may increase the risks of snakebites. These traditions continue to cause snakebite envenomings and deaths in remote areas of Nepal.

Unlike the findings in a cross-sectional survey in villages of 23 districts of the Terai region of Nepal [6], such envenomings occurred mainly while asleep like in envenomed cases from next study performed in far- and mid-western Nepal [11]. Although the recall bias in community-based surveys might cause this variation, our findings correspond to the distribution and diversity of nocturnal medically relevant snake species distributed in Nepal [10,11,25,27].

Similar to a contemporary, hospital-based study in western Nepal [11], we found the next highest envenomings that occurred under natural conditions during agricultural activities by stepping on or placing the hands or legs near an unseen snake. Envenoming occurred while intentional human-snake interactions (i.e., during attempts to keep them away from populated areas). Interestingly, similar to a case revenging against a venomous snake, it was also exhibited by a person inhabiting Ajnawa village of Mahisagar District in Gujrat of India (https://www.onlinekhabar.com/2019/07/782014). Overall, the risky places and activities vary depending on the involved species of snakes [50], noticeably greater number of envenomings during sleeping and agricultural and labor-intensive activities than other activities of victims (p-value: <0.001, FET) also suggested need of an educational interventions ensuring safety from snakebite while sleeping and agricultural activities in all snakebite affected districts of Nepal.

Unlike previous studies [6,10], we found upper extremities being mainly bitten (Table 2) particularly while victims were active (Table 3). A study of envenomated cases representing far- and mid-western Nepal also observed the majority of cases (i.e., 40%, n = 57) being bitten in the upper extremities [11]. But, there was an exposure of lower extremities of victims when bitten by kraits and Russell's Viper that is distributed along the Terai of Nepal [10]. Overall, the snake bitten upper and lower extremities suggest that extremities disturbed snakes which in turn caused snakebites. Further, we found 58% of the envenomings occurring on extremities during victims' active status which could expose their body parts to the venomous snakes.

The highest risks of envenomings occur in July and at night in Nepal (Table 2). Similar risks for envenomings were also reported in other fragmentary studies in this country [10,13]. A study of snakebite envenomings based on Bheri Hospital in western Nepal reported the peak of envenomings in September, but at night [11]. This suggests that nights of July to September are noticeably at high risks for snakebite envenomings mainly while people are at passive conditions (Table 3) and in human-inhabited areas (Table 4). Except during day hours, the envenomings reported in other periods occurred predominantly in human-inhabited areas (Table 4). Similarly, envenomings that occurred at night happened while people were inactive (Table 3). Our findings of most of envenomings during dark hours of the day in monsoonal months while people were inactive and using residential areas correspond to increased activities of snakes for foods and breeding and people for crop cultivation and harvest.

The majority of cases being envenomed by cobras (p-value: <0.001, Table 2) correspond to the highest frequency of cobras known from a study of medically relevant snakes based on nine STCs in Nepal [10] and from snake-photo album displayed to surveyed participants in the Terai of Nepal [6]. However, the fragmentary studies of snakebite envenoming carried out at Bharatpur Hospital in the south-central [43] and Bheri Hospital in the south-western Terai of Nepal [11] reported kraits (*Bungarus* spp.) to be the most common cause of envenoming in the service areas of respective hospitals. However, from south-eastern Nepal, cobra bites are more common than krait bites [25]. However, bites due to these snakes were independent of activeness and in and out of human habitations used by victims (Tables 3 and 4). There are proven envenomings due to pitviper species bites in the hilly districts of Nepal [a Mountain Pitviper (*O. monticola*) envenoming in Kathmandu [30], a White-Lipped Green Pit Viper (*T. albolabris*) envenoming in Gorkha [31], pitviper (species not mentioned) envenomings in Achham [51]]. Further, envenomings due to Greater Black Krait (*B. niger*) at 1515 m asl in Ilam District [29] as well as our findings of several fatalities in the hilly districts (Fig 5) suggest the urgent need of multi-center and large-scale study on preserved snakes brought by bitten patients or their visitors to know the variation in the composition of species involved in envenoming attitudinally and in the eastern, central, and western Nepal. Additional studies on geographical distribution patterns and movement behavior of these medically highly relevant snakes [10,22] can give more detailed ideas on their involvement in snakebites.

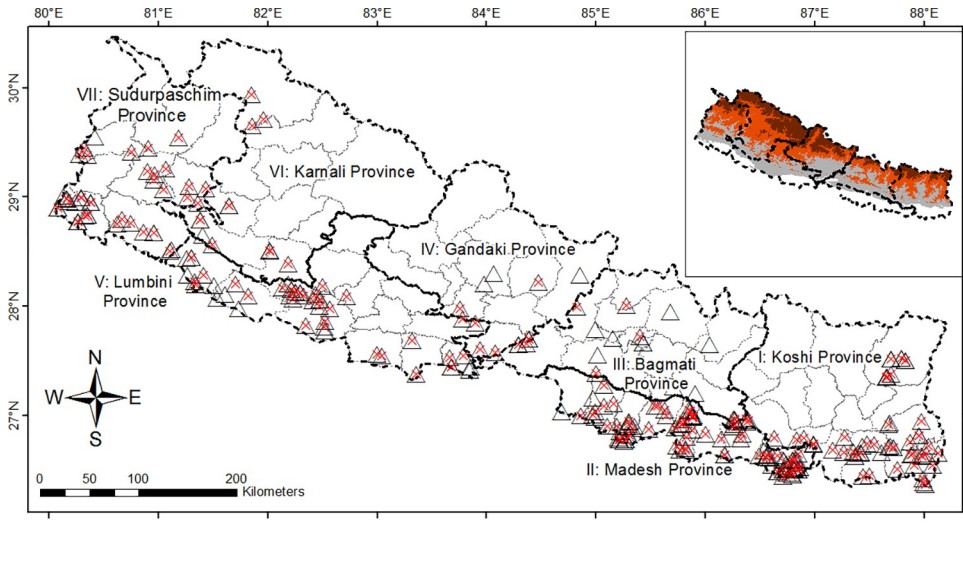

**Snakebite case reports**        **Political divisions**

× Deaths (n = 229)              ⬚ Districts

△ Envenomings (n = 308)        ⬚ Province

**Fig 5. Geographic locations of the news media-reported snakebite envenomings and deaths with substantial information [Locations of envenomed cases are shown in black open triangles and localities from where deaths were reported are indicated with red "X" symbols].** The details on sources of envenomings and deaths are mentioned in Table 1, S3 Table, and S1 Data. The coordinates of mentioned locations are in data file (i.e., S1Data). [The first author of this study created this map in ArcGIS 10.1. The source of the basemap shapefile onto which data has been plotted was used from an openly available source (https://gadm.org/data.html)].

The media reports of snakebite envenoming are increasing in Nepal. Therefore, the biases in snakebite envenomings reported by journalists are likely due to unseen snakes. Similar to news media-reported snakebite analyses in the USA [17], our findings of 228 envenomings without the type of snakes (Fig 2) indicated that the majority of envenomings from unseen snakes appear to be the norm in Nepal. Therefore, evidence-based information on snakes and their ecology is essential to understand the precise distribution patterns and movement of medically relevant snake species involved in envenomings elsewhere. Snake population studies may also give ideas on whether snakebite envenomings correlated to the populations of the medically relevant snake species. Because of snakes' ecological roles and medicinal values [21], eradication of snakes is impractical. Therefore, the pragmatic educational interventions targeted to snakebite-prone regions should include the aforementioned risks factors of snakebite envenomings.

## iii) Consequences of prehospital intervention (DOH) and treatment-seeking behavior

Delayed hospital admission requiring a median of 3.3 h is still a challenge for snakebite management in Nepal. The consultation of traditional snakebite healers (TSHs) by many snakebite victims instead of seeking antivenom therapy directly (Table 5) delayed the hospital admission keeping them at the risk of death. Usually, TSHs apply tourniquets, incision of bitten body part and sucking of blood, chanting, and use of herbal and non-herbals [13]. For the appropriate care, World Health Organization recommends a maximum travel time limit of 1 h to reach healthcare facilities supplied with antivenom and ventilators [52], although this duration varies

greatly depending on the venom effects of snake species and mode of transport. A hospital-based study of envenomings in south-western Nepal reported 0.3 to 95 h (median 5 h, IQR: 2–14 h) required to reach a qualified healthcare facility [11]. Therefore, Nepal requires mass awareness to increase the timely admission of envenomed patients in a STC nearest to the areas where snakebite occur and minimize fatalities due to delayed antivenom treatment.

The dependency on traditional healers is still commonplace in the Terai and the hills of Nepal. A recent study of snakebite traditional healers in eight districts of Nepal [13] mentioned the ingrained faith of people in traditional healing, unaffordable modern care, and wishes for early treatment of snakebites to be the leading causes of this dependency. Similar consultations of TSHs by snakebite envenomed patients were reported in a Ph.D. dissertation carried out in Nepal (i.e., 15%) [10]. Seeking treatment for snakebites from TSHs exposes patients to useless and/or non-recommended interventions for snakebite management. Therefore, consulting TSHs is a major cause of delay in receiving definitive care for snakebite in Nepal [13,37,38,43,53]. However, consulting TSHs for treatments of snakebites is less compared to the past when hospitals were further away and roads and transport networks were inadequate. This might cause no association of consulting TSHs on outcomes (Table 5) although the treatment seeking at a healthcare facilities was associated with consultations of victims with TSHs (p-value = 0.001, PCTI). Compared to the past reports, treatment-seeking behavior of people bitten by snakes in Nepal has been improved. However, reliance on TSHs for snakebite treatment is still a major challenge for snakebite management in this country.

This dependency exists elsewhere in other countries in Asia [Vietnam [54], Myanmar [55], India [56], Sri Lanka [50]] and Africa [Ghana [57], Kenya [58,59]] where it contributes to increasing snakebite fatality rates. Therefore, mass awareness campaigns and trainings on proper prehospital care are essential to ensure immediate access to a snakebite treatment centre in areas where people mostly rely on TSHs for snakebite care. This approach can diminish the dependency on TSHs, help to standardize the health of rural populations [37], and can play a vital role in reducing the number of fatalities resulting from venomous snakebites. Further, additional causes of this dependency [13] should be identified and addressed accordingly to improve the treatment-seeking behavior of people inhabiting snakebite-prone regions elsewhere.

### iv) Outcomes

The geographical locations of reported deaths (Fig 5) signify the snakebite hotspots in Terai to Mahabharat ranges of Nepal for the first time. These deaths more frequently occurred in tropical regions than in subtropical and temperate regions [p-value: < 0.001 (PCTGF)]. The report of snakebite deaths from some mountain and hilly districts, however, raised more serious concerns of managing snakebites nationwide. Significantly large number of deaths (55%, n = 106) occurred before reaching healthcare systems [p-value: <0.001 (PCTGF)] i.e., on the way to a hospital, at homes, at a traditional healer's homes, and Chhaupadi huts. Our findings of dependency of outcomes on modern treatment seeking of victims (p-value = 0.002, Table 5) support the most of out-of-hospital deaths. Seeking traditional healing prior to antivenom therapy, referral of cases to distantly located healthcare centers, inadequate roads and transport networks, and ignorance of people about the snakebite envenoming and STCs nearby their activity areas delayed timely receiving of antivenom therapy resulting in deaths. Similar consequences were reported in hospital-based studies of envenoming in several fragmentary studies (south-central [43] and western Nepal [11,60] and community-based study in south-eastern Nepal [38]). The majority of out-hospital deaths are reported in India, too [61]. However, the majority of deaths at STCs [n = 92, (47%)] indicated improvement in treatment-

seeking behavior of people bitten by snakes in Nepal. Nearly two decades ago, 40% of deaths occurred in the village, 40% on the way to the hospitals, and only 20% occurred in the hospitals in south-eastern Nepal [38]. The recent hospital-based study carried out in south-western Nepal showed 60% prehospital deaths and 40% deaths at the hospital during treatment [11]. Although there is improvement in treatment-seeking behavior in Nepal, adoption of the dual care systems (Table 5) can be detrimental. Such dilemma is prevalent in Nepal [13] and elsewhere [62]. To measure the declining influence of TSHs on snakebite care, there is a need of analyzing community- and hospital-based snakebites altogether [56]. This depicts the whole gamut of burden due to snakebite envenomings, prehospital care practices, and associated consequences.

An envenemed patient referred from Taplejung Hospital reached the snakebite treatment centre at Charali of Jhapa District on the third day of the snakebite. By that time, his condition was serious. Hence, he was referred to BP Koirala Institute of Health Sciences, Dharan, Sunsari District, where doctors managed to save him by amputating the bitten finger (Fig 2). Further, referred cases often died on the way because the higher healthcare centers were far away from the original centers. So, if cases are going to be referred to a higher center located far away from the existing center, all necessary medications and ventilation support should be arranged adequately. The frequency of medical attention after snakebite (Table 5) suggests a need for multifaceted community health education programs [63,64] to expedite patients' follow-up to the antivenom therapy in a timely manner which increases the percentage of survival/recovery and decreases fatalities. Children's carelessness and inadequate knowledge about snake and snakebite [65] expose them to human-snake conflicts [21] resulting in more envenomings and subsequent deaths. To understand the greater mortality of children than adults and elders, there is a need of more sophisticated study on comparison of the effects of to coagulotoxic and neurotoxic snake venoms among envenemed minors and adults [66]. However, considering larger number of minors' deaths and greater proportions of injected venom per kg body weight, pediatric intensive care units should be established in STCs at an accessible distance from the origin of snakebites to ensure treatment on time.

Like in India [61], the majority of deaths occurred during the rainy season (Table 5) and in the tropical Terai of Nepal. More snakebite envenomings and associated deaths occurred in tropical region than in other regions could be associated to the greater diversity of highly venomous and medically relevant elapid snake species (Tables 2 and 5, Fig 5) and their population density in the tropical region of Nepal. Although outcomes of envenomings were independent of remoteness of snakebites in Nepal (Table 5), the greater proportion of snakebite envenomings and associated deaths in urban and semi-urban areas of Nepal indicated populations inhabiting non-rural areas at the risk of snakebite envenomings and deaths, too. However, more sophisticated study is needed to confirm these associations. Envenoming to sleeping people delays treatment compared to those who are alert because krait bites are often painless, snakes are invisible, and victims know effect of venom late. Consequently, the delay in proper treatment causes fatalities. Therefore, apparent envenoming in sleeping people should be considered seriously and carried to the nearest STC as soon as possible.

The majority of deaths (i.e., 43%) were caused by Russell's Viper (*Daboia russelii*) in India [61], whereas cobras (*Naja* spp.) followed by kraits (*Bungarus* spp.) were responsible for the majority of deaths in Nepal. Although the snake types involved were not associated to the outcomes, the more deaths of victims who were passive during snakebite (Table 5) suggested envenoming by intruded nocturnal elapids (herein, kraits) into residential areas in search of prey animals [27]. Therefore, knowing the movement ecology and food habits of these snakes in snakebite-prone zones (Fig 5) is essential to evaluate any association of snake ecology with patterns of envenomings, and with the behaviour of people engaged in different activities

while snakebites. This helps to predict high-risk factors for snakebite envenomings and develop pragmatic prevention strategies against cobras (*Naja naja*, *Naja* spp.), kraits (*Bungarus caeruleus*, *B. fasciatus*, *Bungarus* spp.), true viper (*Daboia russelii*), and pitvipers (*Ovophis monticola*, and *Trimeresurus* spp.) which often cause envenomings and deaths in Nepal.

Because the proportions of deaths depend on other multiple causes (Table 5), the duration between time of snakebite and declaration of death of envenomated patients was highly variable like the length of time between snakebite and initial noticeable symptoms [in an elapid snakebite from Bharatpur Metropolitan City of Nepal [43], ptosis developed after 26 h of snkaebite]. However, public misconception of immediate deaths after snakebites [13,65,67] can be minimized referring to our findings of the noticeably greater median length of time (i.e., 6 h) between snakebite and declaration of deaths. This helps to pacify the envenomed victims. Notably, the study of situations and activities of snakes in ecosystems where community people frequently interact with snakes, public perceptions on snakes [21,67], and knowledge of people on the availability of antivenom treatment facilities within an accessible distance from the origin of snakebites should be integrated to develop evidence-based control measures and minimize the snakebite deaths and meet the World Health Organization's goal of halving snakebite related deaths and disabilities in the South-East Asia by 2030 [68].

### v. Limitations

Our search could collect only a subset of all envenomed cases from Nepal. Follow-up reports of the envenomed cases were not available. So, we were unable to report the outcomes of the cases being under treatment. Many snakebites occurred in the far-flung villages might have yet to be reported in the news media included in this study. Similarly, it was also possible that we might have yet to come across some of the reports because of search keys and engines that we used retrospectively. Many factors not available in this study might confound effective treatment and outcomes. Searching news media data prospectively by using additional search strategies can increase case reports. We primarily accessed newspapers. Searching of snakebite issues announced by televisions and radios may further increase the snakebite case reports. Further, encouragement of journalists to publish detailed snakebite case reports including clearly defined age, sex, occupations, means of transport used, dosage of antivenom administered, length of time between snakebite and hospital access, discharge, death, and referral (if any) of victims' to higher healthcare center, authentically identified venomous snake types, etc. increases the opportunity for more powerful statistical analyses by prospective researchers.

### Conclusions

Snakebite envenomings mostly affected children and both males and females had equal probability of being envenomed due to venomous snakebites in Nepal. These envenomings more frequently occurred on upper extremities due to bites of cobras followed by kraits. Residential areas at night and monsoonal months (July followed by August and June) during sleeping are at risk of envenomings. Noticeable fatalities of envenomed victims while sleeping suggest the urgent need of formulating pragmatic and research based prevention and prehospital care strategies. Properly designed community-based educational interventions by integrating governmental institutions (e.g., hospitals, schools, etc.), epidemiologists, clinicians, and toxinologists, serpentologists, and community-based non-governmental organizations are essential to spread measures for snakebite prevention (e.g., improvement of sleeping behavior of people (by encouraging to sleep on cot-bed using mosquito net and to examine bed-sheet and pillow before each bedtime and discouraging to sleep on floor-bed) and their housing with screened doors and windows to keep snake prey animals away from houses, etc.) and prehospital care.

The dependency on TSHs for snakebite treatment is still a major challenge for snakebite management in this country. Therefore, educating mass people about venom effects (i.e., evolution of symptoms) and its consequences (deaths and morbidity) can motivate people to seek appropriate treatment timely. For the further attraction towards modern care of snakebite patients, Nepal Government should establish health insurance policies covering travel and treatment costs for the impoverished snakebite victims. Subsequently, carrying people to the hospital in a timely manner after snakebite mitigates snakebite management challenges, thus increasing survival and decreasing fatalities. Several snakebite deaths known from hills and mountains suggest an urgent need of snakebite treatment facilities. Therefore, prevention and treatment strategies should include entire Terai region and some mountain and hilly districts of Nepal to reduce snakebite fatality substantially. Further, the Nepal Government should update the envenoming risk map to ensure effective and efficient management of snakebites nationwide. Our findings can be used to design more representative epidemiological study to precisely extrapolate snakebite burden by considering the epidemiological situations in snakebite prone, tropical urban and semi-urban areas of Nepal's Terai. Further, to reduce fatalities and morbidities associated to snakebites, our epidemiological mapping of snakebite envenomings can be used to protect populations at-risk of envenomings and deaths, particularly children, and carry out more effective interventions nationwide and elsewhere having similar geo-climatic conditions and socio-economic status of people, particularly in countries where the government has no interest in registering cases of snakebite envenomings.

## Supporting information

**S1 Table. Names and languages used in the news media and URLs of articles included in this study.**
(DOCX)

**S2 Table. Incidence of snakebites and associated envenomings and deaths reported in the news media during 2010–2022.**
(DOCX)

**S3 Table. Comparison of distribution of snakebite envenomings and associated deaths reported by the news media during 2010–2022 by districts (and institutions) with a published report of similar incidence from 23 districts of Terai region of Nepal.**
(DOCX)

**S1 Data. Data file used to analyses of this study.**
(TXT)

## Acknowledgments

We would like to thank Gita Subedi Pandey for her tireless support to search, find, and compile news media-reported snakebites and data entry coded by authors. Further, we are grateful to Prof. Elda E. Sánchez from National Natural Toxins Research Center, Texas A&M University, USA and anonymous reviewers for their constructive comments.

## Author Contributions

**Conceptualization:** Deb P. Pandey.

**Data curation:** Deb P. Pandey.

**Formal analysis:** Deb P. Pandey.

**Investigation:** Deb P. Pandey, Narayan B. Thapa.

**Methodology:** Deb P. Pandey.

**Project administration:** Narayan B. Thapa.

**Resources:** Narayan B. Thapa.

**Supervision:** Deb P. Pandey.

**Validation:** Deb P. Pandey.

**Writing – original draft:** Deb P. Pandey.

**Writing – review & editing:** Deb P. Pandey, Narayan B. Thapa.

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
