## [Decision Letter · Decision Letter 0]

10 Jul 2023

Dear Dr. PANDEY,

Thank you very much for submitting your manuscript "Analysis of News Media-Reported Snakebite Envenoming in Nepal during 2010–2022" for consideration at PLOS Neglected Tropical Diseases. As with all papers reviewed by the journal, your manuscript was reviewed by members of the editorial board and by several independent reviewers. In light of the reviews (below this email), we would like to invite the resubmission of a significantly-revised version that takes into account the reviewers' comments. 

We cannot make any decision about publication until we have seen the revised manuscript and your response to the reviewers' comments. Your revised manuscript is also likely to be sent to reviewers for further evaluation.

Sincerely,

Wuelton M. Monteiro, Ph.D.

Section Editor

Wuelton Monteiro

Section Editor

Reviewer's Responses to Questions

**Key Review Criteria Required for Acceptance?**

**Methods**

-Are the objectives of the study clearly articulated with a clear testable hypothesis stated?

-Is the study design appropriate to address the stated objectives?

-Is the population clearly described and appropriate for the hypothesis being tested?

-Is the sample size sufficient to ensure adequate power to address the hypothesis being tested?

-Were correct statistical analysis used to support conclusions?

-Are there concerns about ethical or regulatory requirements being met?

Reviewer #1: The study design was appropriate to achieve the proposed objective.

The approach in which this article addresses the epidemiology of venomous snake occurs in Nepal is one of its primary contributions. The authors conducted a comprehensive evaluation of data from reports and mapped reported cases across the country. This provided an in-depth comprehension of the geographical distribution of venomous snake bites and the demographic characteristics of the victims. The finding that children are the most vulnerable group is alarming, but it also highlights the significance of focusing preventive efforts on this age group. In addition, this methodology can be applied to additional contexts and countries in future research. Especially in countries where the government has no interest in registering cases of snake venom poisoning.

Reviewer #2: 1.

NO, IT IS JUST DESCRIPTIVE. THERE IS NO STATISTICAL ANALYSIS AND TEST HYPOTHESIS AT ALL.

2.

Yes, But there is a need for improvement.

how do the authors deal with the Data to achieve stated objectives?

what they do as descriptive is not enough to address the stated objectives.

but If the authors have a point of view about the research objectives How was it achieved? Please Explain that.

3.

Yes, the population is clearly described But there are no hypotheses being tested.

4.

No. Please provide more details, and Explain that.

Did the authors calculate the sample size?

The size of the sample is 296 published media news. What about the unpublished snakebite incidents?

Which can be registered with hospitals or another way? How was it taken into account?

The sample size is 296, but during research paragraphs, you used several other numbers!!

such as the number of envenoming cases=308 

How do you justify that the number of envenoming cases is greater than the sample size?

Why did the sample not adopt the number of envenoming cases instead of the published news?

When calculating the ratios, it was not clear that they belong to the number of envenoming cases or to the number of published news In addition to your use of Figure No. 2, the sample size = 533

Please specify precisely and clearly a fixed sample size.

5.

No.the authors did not use statistical analysis tests. and here is how they should be improved:

A. Enriching the research with statistical tests Not just descriptive narration numbers and percentages. So Each of the following paragraphs requires at least one statistical test for the results to be significant.

Line 260 i) Demographics

Line 275 a) Locality of snakebites

Line 285 b) Places and victims' activities when bitten

Line 293 c) Bitten body parts

Line 299 d) Temporal patterns of snakebite envenomings

Line 304 e) Snakes responsible for bites

Line 313 (DOH) and treatment-seeking behavior

B. Determining the factors affecting snakebite incidents through chi-square tests that fit the data (independence test and good fitness test), which factors are more influential in the geographical location? The time of the accident? Characteristics of the victim? Age, gender, or others, so multivariate analysis can be used and constitute an added value to the research results.

C. Use the Mean ± SD to describe numeric data, You Used median and IQR Explain Why?

D. I strongly recommend using percentages and the number next to each other, indicating the percentage to whom it belongs, for example, lines 27 - 34. Put the percentages next to each number mentioned and specify whether the percentage is from 296 or is it the percentage from 308

Some paragraphs shifted the number and proportions to this, but some paragraphs did not. Explain that

6.

NO.

Reviewer #3: The strength of this paper is it clearly explains the objectives of the study the way it has written is that anyone who can read the paper can be easily understand it. However it is better to include the statement of the hypothesis clearly to be tested in order to achieve the objectives of the study in the paper .the inclusion and exclusion of the data in the paper is good and clearly understandable for any readers who were out of specialization.

 The study design for this paper of analysis is an clear and attractive. I am looking here is that in the method of data analysis, in search of traditional snakebite healers, there is no clear way which variable is collected from this source of data ( I mean whether they have conduct an interview or they have used publish documents). In addition in this source it is not clear how to determine the species of snakes. 

In the methodology part states that the continuous variable is normally distributed and median, range and inter-quartile range is used for the analysis. If there is a continuous variable arithmetic mean, variance, standard deviation, minimum and maximum values are appropriate to describe the output of the result.

The population that have defined in the paper is clearly described and appropriate for the research objectives. The sample size in the paper for which organized is so much enough to made adequate conclusion or decisions for the hypothesis. 

The sample is already taken from the media reports; there is no any procedure to select the sample from the population. Hence it is adequate to test the hypothesis of the research objectives.

 The statistical analysis is good to support the conclusion. But you have not test the normality of the variable. Since the organization of the paper uses online public data, there is no ethical concern.

Reviewer #4: The study's main objectives have been outlined, but there is a need for clarity on some of the study variables. Specifically, in section 217, "Circumstances" is categorised as occurring inside or outside human habitations. However, previous statements in sections 181-182 indicate that snake bites' time, month, and year should also be considered under the "Circumstances." While the study results include some of this information, clearly stating the objectives would make it easier for readers to understand. The study needed more clarity in defining the population regarding district borders (81) and stating the population count of inhabitants in the area but rather provided information on the entire area's climatic changes (120-124). The sample size was sufficient, although it is important to note that the age grouping should be systematically organised to ensure regular age intervals (214).

This study retrospectively analysed existing public data and did not need ethical approval (117). The study included hospital-reported cases to gather information about treatment outcomes, which was not disclosed as a data source (160-168). Additionally, using patients' names was mentioned at 189, raising concerns about data protection that should be addressed ethically (200-203). In order to ensure the study is replicable, it is important to mention the specific news outlets from which the data was sourced. The attached PRISMA flow diagram did not provide this information, even though the article states that data was obtained from Google, Newspapers, Facebook, and other sources. 

It is important to clarify which data came from which sources. While the study mentioned accessing data from treatment centres and Google, according to pages 566-567, it primarily came from newspapers, indicating a contradiction. The data analysis was attributed to Microsoft Excel, but further reading reveals that data was also analysed using Google Earth Pro and R statistical programs (213 and 249).

Reviewer #5: Accept

**Results**

-Does the analysis presented match the analysis plan?

-Are the results clearly and completely presented?

-Are the figures (Tables, Images) of sufficient quality for clarity?

Reviewer #1: The results presented by the authors corroborate to achieve the objective proposed in the manuscript.

It is also possible to verify which population presents the highest risk, which snake species is the major cause of accidents, and the circumstances of the ophidian accident.

Reviewer #2: 1.

NO, IT IS JUST DESCRIPTIVE. THERE IS NO STATISTICAL ANALYSIS AT ALL.

2.

As descriptive, yes. but there is a need for improvement and here is how they should be improved:

THE ANALYSIS RESULTS WE CAN NOT DEPEND ON WITHOUT SIGNIFICANT STATISTICAL TEST WITH P-VALUE.

Note that: Based on your research data, if the authors say that the percentage of envenomated women is higher than that of men, Because women are 160 (52%) and men are 144 (47%).

This is not considered a result of scientific research, it is only considered a descriptive statistic from a sample, because the difference between the ratios by 5% has not been tested statistically, So this difference may be NOT significant, until it is verified by testing. (you must use Chi-square test).

However, the authors dealt with it and other descriptive statistics as proven scientific results, and this is a grave mistake.

3.

The graph and maps are good and have an important clearance for all the information. 

Is it possible to choose a higher resolution? For clarity without pixelation

Reviewer #3: The analysis presented in the paper very good and it is perfectly conceited with the analysis plan settled in the organization of this paper. But one variables of interest has been missed in the analysis (occupation of snakebite)

The result of the analysis is clearly presented. In almost all it is good and clear for any reader from other specialization to understand it. However, Not Applicable (NA) data values must be replaced by appropriate values. Since NA values are missing data values, it is better to treat missing data values with appropriate missing data treatment methods. Even if the result for the analysis of the paper is almost all completed for all variables described in the methodology and its way of presentation is good, for the variable occupation of snakebite cases (farmer, student,. . .) is not listed in the result section. In the result section, age and hours must be described in terms of average, standard deviation and variance instead of median, range and inter-quartile range.

I have given a great credit for the way of presenting figures and images put under the appendix part for the output. The output of figures (tables and images) displayed in the appendix part of the results for the analysis data is clear and sufficient for the describing the output of the analysis for each variable of interest. The quality of tables and images are enough to get clarity for the results describe in the result section. Abbreviations under subscript of each table have been clearly defined. Generally it is good, clear and sufficient.

Reviewer #4: The study's main objectives were achieved in the results, but some modifications are necessary to ensure the article is well organised and flows smoothly.

Reviewer #5: The results are not clearly and completely presented. Because at line 260 (Demographics) through to 264 the statement made doesn't correspond to the table 2. It was stated that "We determined that 262 sex was known in 304 cases but without age in 59 cases". So the issue here is the 59 cases without age. Where from it and if you can explain it better or make corrections.

**Conclusions**

-Are the conclusions supported by the data presented?

-Are the limitations of analysis clearly described?

-Do the authors discuss how these data can be helpful to advance our understanding of the topic under study?

-Is public health relevance addressed?

Reviewer #1: This study proposes methods for mitigating the effects of venomous snake injuries. Using the collected epidemiological data, the researchers identified high-risk regions and established a firm foundation for the development of prevention and intervention programs. 

One suggestion: The authors could expand the discussion on ways to mitigate the occurrence of snakebites, such as the implementation of educational measures, such as awareness campaigns and first aid training. This can play a vital role in reducing the number of fatalities resulting from venomous snake bites.

Reviewer #2: 1.

NO,All conclusions are based on descriptive numbers, and this is not sufficient to be considered a conclusion, because the relationships and differences assumed by the authors may not be statistically significant unless this is proven.

2.

Yes, there is no need for improvement.

3.

Yes. but there is a need for improvement.Because The Scientific research should not be just collecting, describing and summarizing news, because in this case it will become a news publication, but not a scientific research.

4.

Yes, there is no need for improvement.

Reviewer #3: The way of the conclusion in the paper is very good and tries to depend on the data presented in the result section. The conclusion must depend on all results obtained in the analysed data. But not generalize based on all variables of findings. It has concluded by only few study finding results or concluded by only few variables. 

In the organization of the paper the limitation has been listed down. In the limitation part is listed only the unavailable data for the media report reports the cases were under the treatment condition. However, in the data analysis part you have described the inclusion and exclusion criteria. Under media reports some information in the cases may be missed like age or age groups, sex . . . These are the limitations in in conducting this paper. It is better to include all important sources which gives power for the quality of paper but not unable to access it.

 This paper discussed on the understandings of snakebite envenoming cases in the study area regarding to demographic, circumstance of snakebite and outcome of bite causes. This paper show an important figure is that how traditional religious-cultural effect on girls and women in the selected study area. The authors have discussed his findings with other authors that have been given primary attention in paper organization. The way how to discuss his findings with another author’s finding has given an advance uses of understandings of the topic in the study area.

From this paper the public health relevance related to the rescue of snakebite envenoming in the study area has been addressed. The authors have fond or determine the categories of society which have been exposed to the topic of the study. Therefor any concerned body can take an action or intervention by understanding this finding in order to protect the society.

Reviewer #4: Carefully analysed data support the study's conclusions, and the study's limitations are clearly stated. The study's benefits are outlined and are especially relevant to the Nepalese, whose daily lives and cultural practices often involve exposure to venomous snakes. While the study successfully answers the questions it raises and achieves its objectives, these objectives were scattered throughout the article instead of being clearly stated in the introduction, which may make it difficult for readers to understand the study's main goals, as noted in section 124.

Reviewer #5: (No Response)

**Editorial and Data Presentation Modifications?**

Reviewer #1: Accept

Reviewer #2: The graph and maps are good and have an important clearance for all the information. 

Is it possible to choose a higher resolution? For clarity without pixelation.

Reviewer #3: In over all this paper is original and it is higher power of significance for other researchers for further investigation and policy makers to revise their policies. It requires minor revision in the abstract; it must include major variables which were having a significant difference findings. In the result part it is better to test the normality test of continuous variables and it must be present the data analysis output in terms of averages or means, variance, standard deviation, minimum and maximum values special for age and duration hours. In the introduction, it is better to include some significant variables.

In methodology it requires to replace missing values with appropriate missing data treatment techniques. It is necessary to include the method of data collection from traditional snakebite treatment healers. It must clear how to identify species of snakes in traditional snake bite healers. It is best to remove useless in formation like inadequate transports which is not found in the data analysis outputs. In the discussion part it must be necessary to discuss for all variables of interest.

It requires minor revision!!

Reviewer #4: The study needs minor revisions to address the issues in the general comment and methodology section.

Reviewer #5: Accept

**Summary and General Comments**

Reviewer #1: The manuscript presents an innovative study that casts light on a critical public health issue in Nepal involving the incidence of venomous snake bites, which result in an alarming number of annual deaths. Nonetheless, this study emphasizes the significance of epidemiological mapping as an effective strategy for reducing fatalities and protecting at-risk populations, particularly children.

Reviewer #2: - Major and minor points and Additional tables must be added.

Information has been provided in the "Attachments" Section.

Last Comment:

-The published media news is influenced by lies, exaggeration and rumors. How do you verify that?

Please provide more details, and Explain that.

Reviewer #3: The strength of the study, is it is noble, highly significant, and the methodology of data collecting strategies, I mean how to data to be included and excluded for the analysis is very interesting and appreciated. Ii also very good in abstract format even if it requires minimum revision the way it has structured is get high credit. But it requires to include more significant outputs from the data analysis either some discussion or conclusion part. in the output part is is so attractive and clear for any one who reads this paper. But is important to test the normality of the continuous variables in the data analysis since in the methodology part it talks about that continuous variables are not normally distributed but not exist in the result part. All the occupation of snakebite variable must be describe in the result section. other wise it is so attractive in clear and sufficient to test the objectives of the study. in the discussion part it is so great the way that the result has been discussed with other authors. it is so good more over to discuss for all study variables of interest like the length of times between snakebite and declaration of deaths.

In conclusion part is important for based on all significant variables. otherwise it is so nice. finally you have cited for all references and I want to appreciate for this even if few references are outdated.

Reviewer #4: The study mentioned at the beginning of the article was conducted in a specific region of Nepal called Terai, but the title needs to reflect this information (110-113). Several statements are difficult to understand throughout the article and contain grammatical errors. The sentence structures in these statements need to be reviewed. Grammatical errors and hard-to-read sentences can be found in various statement numbers, including 23-24, 30-32, 50-51, 71-74, 112-113, 367-370, 387-391, 398-399, 437-438, and 527-529. Conducting a thorough review of the article's grammar would be appropriate. The word "useless" in statement 42 is offensive, and a more appropriate term should be used. Statement 91-92 is insignificant unless the author wants to highlight the importance of obtaining data from these media outlets.

The authors want to provide the biological classification and common names of the snakes under study. However, their use should be clearly stated to avoid confusion. There should be consistency in the use of common names or biological classification for easy understanding, particularly for non-experts in the field of study. These inconsistencies can be found in statements 39, 55, 133-135 and some parts of the article. 

Using synonyms for "envenom" and "novel" would make the article more enjoyable, as these words are repeated frequently. Although they may be keywords used in the search strategy, consider using other words with similar meanings to interest readers. Some statements are out of place and should be mentioned under the inclusion and exclusion criteria (197, 208-209).

Reviewer #5: Participant with insufficient data could have been excluded from the study (eg. those without age, sex and other vital information)

Moreover taking data from newspapers, television and radio could cause duplicates because one info can be channeled through all the three medium

PLOS authors have the option to publish the peer review history of their article (what does this mean?). If published, this will include your full peer review and any attached files.

Reviewer #1: Yes: Raul Afonso Pommer-Barbosa

Reviewer #2: Yes: Imad Addin Almasri

Reviewer #3: Yes: Asmare Adane Mekonen

Reviewer #4: No

Reviewer #5: Yes: Manfred Dakorah Asiedu
---

## [Editor Report · Decision Letter 1]

4 Aug 2023

Dear Dr. PANDEY,

We are pleased to inform you that your manuscript 'Analysis of News Media-Reported Snakebite Envenoming in Nepal during 2010–2022' has been provisionally accepted for publication in PLOS Neglected Tropical Diseases.

Best regards,

Wuelton M. Monteiro, Ph.D.

Section Editor

Wuelton Monteiro

Section Editor

---

## [Editor Report · Acceptance letter]

20 Aug 2023

Dear Dr. PANDEY,

We are delighted to inform you that your manuscript, "Analysis of News Media-Reported Snakebite Envenoming in Nepal during 2010–2022," has been formally accepted for publication in PLOS Neglected Tropical Diseases.

Best regards,

Shaden Kamhawi

co-Editor-in-Chief

Paul Brindley

co-Editor-in-Chief
